# AgentMD: Empowering language agents for risk prediction with large-scale clinical tool learning

Qiao Jin [1], Zhizheng Wang [1], Yifan Yang [1,2], Qingqing Zhu[1], Donald Wright [3], Thomas Huang [3], Nikhil Khandekar[1], Nicholas Wan[1], Xuguang Ai[3], W. John Wilbur[1], Zhe He[1,4], R. Andrew Taylor [3,5], Qingyu Chen [3] & Zhiyong Lu [1] ✉

Clinical calculators play a vital role in healthcare, but their utilization is often hindered by usability and dissemination challenges. We introduce AgentMD, a novel language agent capable of curating and applying clinical calculators across various clinical contexts. As a tool builder, AgentMD first uses PubMed to curate a diverse set of 2,164 executable clinical calculators with over 85% accuracy for quality checks and over 90% pass rate for unit tests. As a tool user, AgentMD autonomously selects and applies the relevant clinical calculators. Our evaluations show that AgentMD significantly outperforms GPT-4 for risk prediction (87.7% vs. 40.9% in accuracy). Results on 698 real-world emergency department notes confirm that AgentMD accurately computes medical risks at an individual level. Moreover, AgentMD can provide population-level insights for institutional risk management. Our study illustrates the capabilities of language agents to curate and utilize clinical calculators for both individual patient care and at-scale healthcare analytics.

Clinical calculators have emerged as indispensable tools in healthcare, providing clinicians with risk assessments for accurate diagnosis and prognostic evaluation[1,2]. As an example, the widely used HEART score[3] assists in evaluating the risk of major adverse cardiac events based on points tallied from History, Electrocardiogram, Age, Risk Factors, and Troponin values, and has been shown to benefit various populations[4]. Despite the success of clinical calculators to enhance efficiency and decision-making in healthcare, their adoption is constrained by several factors. Clinicians must recognize when and how to apply these tools, necessitating extensive knowledge of them – an issue compounded by slow dissemination[5]. Clinical calculators are also frequently viewed as stand-alone tools, rarely combined together or applied at the same time. In addition, the requirements of non-standardized input parameters and poor integration with electronic health records force clinicians to input data manually, interrupting clinical workflows. This

not only hampers efficiency but also raises the risk of data entry errors[6]. Moreover, the subjective interpretation of calculator components contributes to the variability of manual computation, further undermining their overall reliability.

Language agents[7] offer a promising approach to bridging the gap between clinical needs and risk calculators. Also known as artificial intelligence (AI) agents, they are autonomous systems enabled by large language models (LLMs) such as GPT-4[8]. One of the main features of language agents is the capability to use external tools[9], such as search engines[10,11] and domain-specific utilities[12–16]. Furthermore, advanced LLMs can create reusable tools[17,18] for other language agents to use. However, existing language agents predominantly address mathematics and coding tasks, with minimal exploration into healthcare.

We introduce AgentMD, a novel medical language agent framework to address two primary objectives: (1) the automated

[1]Division of Intramural Research (DIR), National Library of Medicine (NLM), National Institutes of Health (NIH), Bethesda, MD, USA. [2]School of Computer Science, University of Maryland, College Park, MD, USA. [3]School of Medicine, Yale University, New Haven, CT, USA. [4]School of Information, Florida State University, Tallahassee, FL, USA. [5]School of Medicine, University of Virginia, Charlottesville, VA, USA. ✉e-mail: zhiyong.lu@nih.gov

 1

curation of a comprehensive library of medical calculators, and (2) the precise selection and application of these calculators to individual patient scenarios. Accordingly, the architecture of AgentMD encompasses two roles: as a tool maker, AgentMD automatically screens PubMed articles to identify and curate relevant risk calculators, culminating in the assembly of a repository of structured risk calculator tools (we name this collection RiskCalcs in this work). In its role as a tool user, AgentMD employs an LLM-agnostic framework capable of selecting, computing, and summarizing the results from suitable risk calculators based on the provided

patient information. Figure 1 shows the architecture overview of AgentMD.

Our evaluations assess both the tool curation and tool-using capabilities of AgentMD. For tool curation, we manually evaluate the generated calculators in RiskCalcs by AgentMD using quality, coverage, and unit test metrics. To evaluate the tool using the capabilities of AgentMD, we apply it to three different cohorts, including a manually curated RiskQA dataset consisting of 350 multiple-choice questions for controlled evaluations; 698 provider notes from the emergency department of Yale Medicine for individual-level evaluations; and 9822

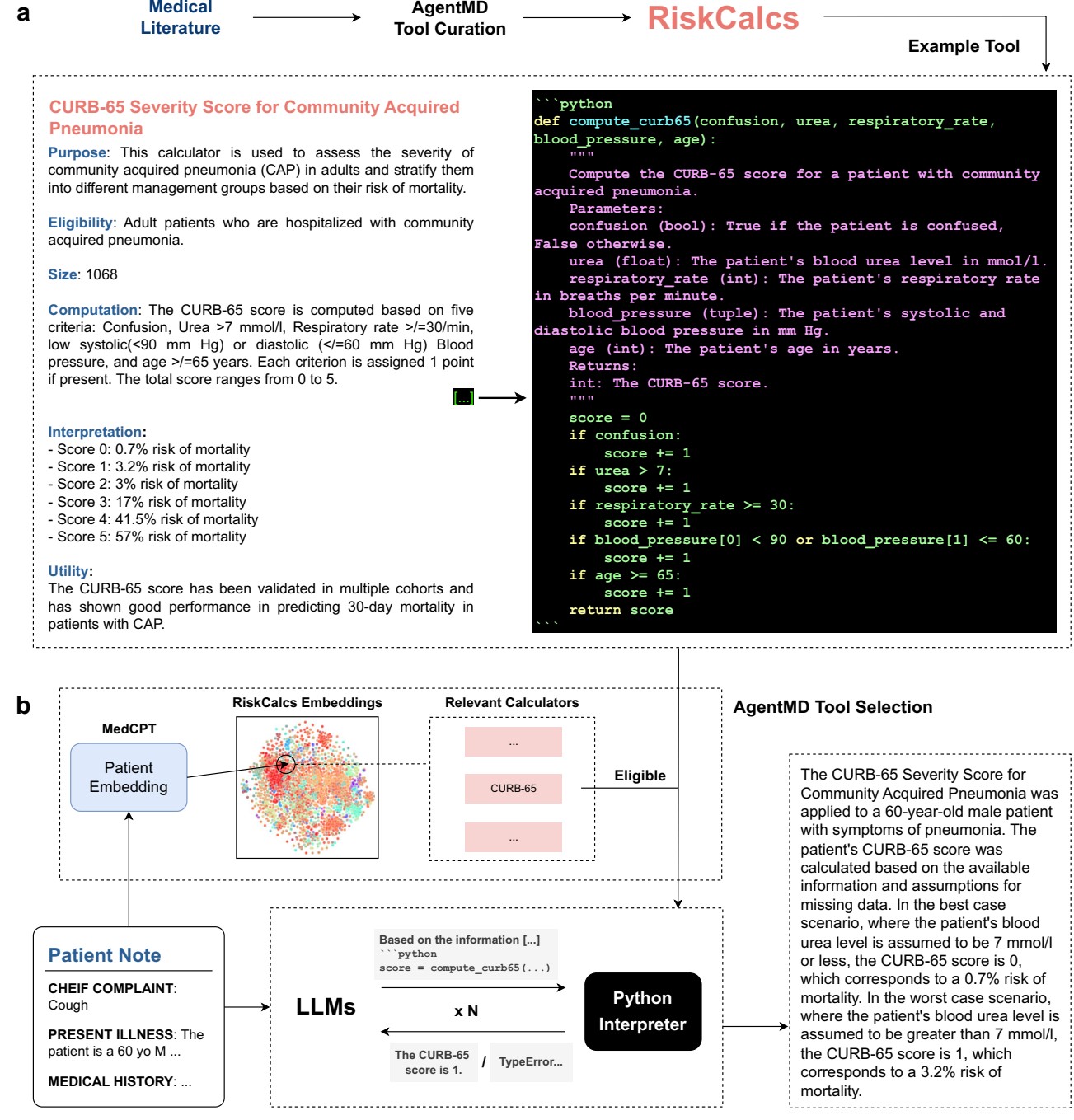

**Fig. 1 | Overview of AgentMD tool curation and using. a** An example clinical calculator in the RiskCalcs toolkit curated by AgentMD, based on the title and abstract of a PubMed article on the CURB-65 risk score (PMID: 12728155). **b** The

methodology of the AgentMD tool using, which includes tool selection, tool computation, and result summarization.

admission notes from MIMIC-III[19] for population-level analytics. Our results demonstrate that AgentMD-curated tools achieve over 85 % accuracy on quality checks and a pass rate exceeding 90 % on unit tests. AgentMD also substantially outperforms GPT-4 on the RiskQA benchmark (87.7 % vs. 40.9 % accuracy), and its effectiveness for both individual- and population-level risk prediction has been validated on two real-world patient cohorts. Although these findings are promising, further and more comprehensive evaluation is required before its practical use in clinical settings.

## Results

### RiskCalcs tools feature high quality and extensive coverage

Figure 2 displays the evaluation results of the calculators in RiskCalcs. We manually evaluate two representative subsets of RiskCalcs: the top 50 most cited calculators (Fig. 2a) and a random sample of 50 other calculators (Fig. 2b). For each calculator, three annotators are employed to evaluate the quality, coverage, and unit test correctness of the tool. The consensus of three annotators is used as the ground-truth labels.

The correctness evaluation includes two aspects: whether the computing logics is correct, and whether the result interpretations are appropriate. Overall, the average correctness of the computing logics and result interpretations are 87.6% and 89.0%, respectively. We also evaluated the pass rate of the unit tests. For this, we used GPT-4 to generate five sets of potential parameter values (Q1–Q5 in Fig. 2) given the computing logics of each risk calculator. We provided AgentMD with both the generated calculator and the parameter set to compute the results, which we denote as AgentMD calculations. Then, we manually performed the result computation with the same set of parameters, either with the raw PubMed abstracts (internal validation) or with the online implementation (external validation) when available. Overall, only 8.4% (42/500) of the AgentMD calculations are inconsistent with manual calculations, while 91.6% of the AgentMD calculations are consistent with manual calculations, respectively. In addition to GPT-4-generated sets, 100 sets of unit test parameters have been manually curated for ten randomly sampled calculators. For this manually curated subset, we specifically included more challenging and edge cases where the patient parameters fall close to the decision boundaries. As a result, AgentMD achieves a passing rate of 84.0%, which is slightly lower but remains overall consistent with the passing rate on the automatically generated unit tests. These results have further validated the accuracy of AgentMD computation and the quality of the RiskCalcs tools.

We also evaluated the coverage of the calculators by checking whether they have been previously implemented as an online tool. For this, we searched the calculators in MDCalc[20], one of the largest hubs of clinical calculators, as well as the first page returned from Google for other online implementations. The majority (68.0%) of the top-25 most cited calculators in RiskCalcs have online implementations. However, the coverage is only 28.0% for calculators between the ranks 25–50. Risk calculators from many highly cited studies, such as the Euro-EWING 99 trial[21], are not implemented by any websites but are automatically converted by AgentMD into a computable tool. Similarly, we did not find any online implementations for most (96.0%) randomly sampled calculators in RiskCalcs. Among the calculators with at least one online version, only 53.8% (14/26) have been implemented by both MDCalcs and other online sources, while the remaining 46.2% have been only implemented in one source. This indicates that manual implementations of clinical calculators are limited in scale and lagging in progress. Overall, our results show that RiskCalcs built by AgentMD can serve as a supplement resource of clinical calculators to the existing online hubs.

### AgentMD can accurately perform risk prediction tasks on the RiskQA benchmark

Unlike the unit tests used to evaluate the curated computing logics by AgentMD, RiskQA is an end-to-end evaluation benchmark that requires

a system to (a) select the suitable risk calculator, (b) conduct correct computing, and (c) provide appropriate interpretations. Experimental results on RiskQA are shown in Fig. 3. When applied to this task, AgentMD first selects a tool from the RiskCalcs collection, then uses it to compute the risk for the given patient and predicts an answer choice, as shown in Fig. 3a.

We first compare AgentMD with Chain-of-Thought (CoT)[22], a widely used prompting strategy for LLMs. AgentMD surpasses CoT by 70.1% (0.546 vs. 0.321 in accuracy, Fig. 3b) and 114.4% (0.877 vs. 0.409, Fig. 3c) with GPT-3.5 and GPT-4 as the base model, respectively. Surprisingly, AgentMD with GPT-3.5 even outperforms CoT with GPT-4 (0.546 vs. 0.409). These results clearly demonstrate that large language models, when provided with a well-curated toolbox of clinical calculators, can accurately select the suitable calculator and effectively perform medical calculation tasks. Figure 3d shows the accuracy of the initial tool selection step. As a baseline, dense retrieval with MedCPT achieves a top-1 accuracy of 0.723. On RiskQA, AgentMD selects the most suitable tool from the top 10 tools returned by MedCPT. Our results show that GPT-4-based AgentMD can better select the required tool than MedCPT, which is in turn better than GPT-3.5-based AgentMD. This highlights the importance of the backbone LLM when AgentMD conducts tool selection. In conclusion, these results demonstrate the effectiveness of AgentMD in selecting and applying clinical calculators as evaluated by USMLE-type questions.

### AgentMD can accurately calculate individual risks from emergency department notes

Emergency care presents pressing challenges as physicians need to provide a comprehensive risk evaluation of the patient in a short period of time. AgentMD has the potential to assist the risk assessment process by automatically selecting and applying medical calculators to inform the risks. To evaluate this use case of AgentMD, three physicians selected a list of 16 commonly used calculators in the emergency department to augment AgentMD, which were applied to 698 provider notes from Yale Medicine (Fig. 4a). For each calculator, two physicians evaluated the top 5 patients that had the most severe risks as ranked by AgentMD. In total, 80 patient-calculator pairs were evaluated. Figure 4b shows the proportions of annotations in each evaluated aspect. Overall, 80.6% of the patients are annotated as eligible for the corresponding calculator, and only 10.6% are annotated as ineligible. Among eligible and partially eligible patient-calculator pairs, over 80% of processes are annotated as either correct (52.3%) or partially correct (28.5%). Within such patient-calculator pairs, almost all calculation results of AgentMD are annotated as either useful (68.6%) or partially useful (29.1%).

Figure 4c shows the per-calculator evaluation results averaged over patients and annotators. Overall, most calculators (14 out of 16, 87.5%) have an average score of over 60%. The only two exceptions with less than 60% overall scores include the HEART Score and Canadian C-Spine Rule (CCR)[23], where the computation process correctness and the result usefulness are scored low due to incorrect assumptions on missing values made by AgentMD. In conclusion, AgentMD demonstrates a high level of accuracy in calculating individual risks from emergency provider notes, with the majority of evaluated patient-calculator pairs showing eligibility, correctness in calculation processes, and clinical usefulness of the results.

### AgentMD can be applied to clinical notes and provide population-level risk insights

In this section, we analyze the population-level risk predictions by AgentMD on the MIMIC-III cohort consisting of 9822 patients. As shown in Fig. 5a, AgentMD first generates a list of potential risks with their quantitative likelihoods for each patient. Then, we aggregate the

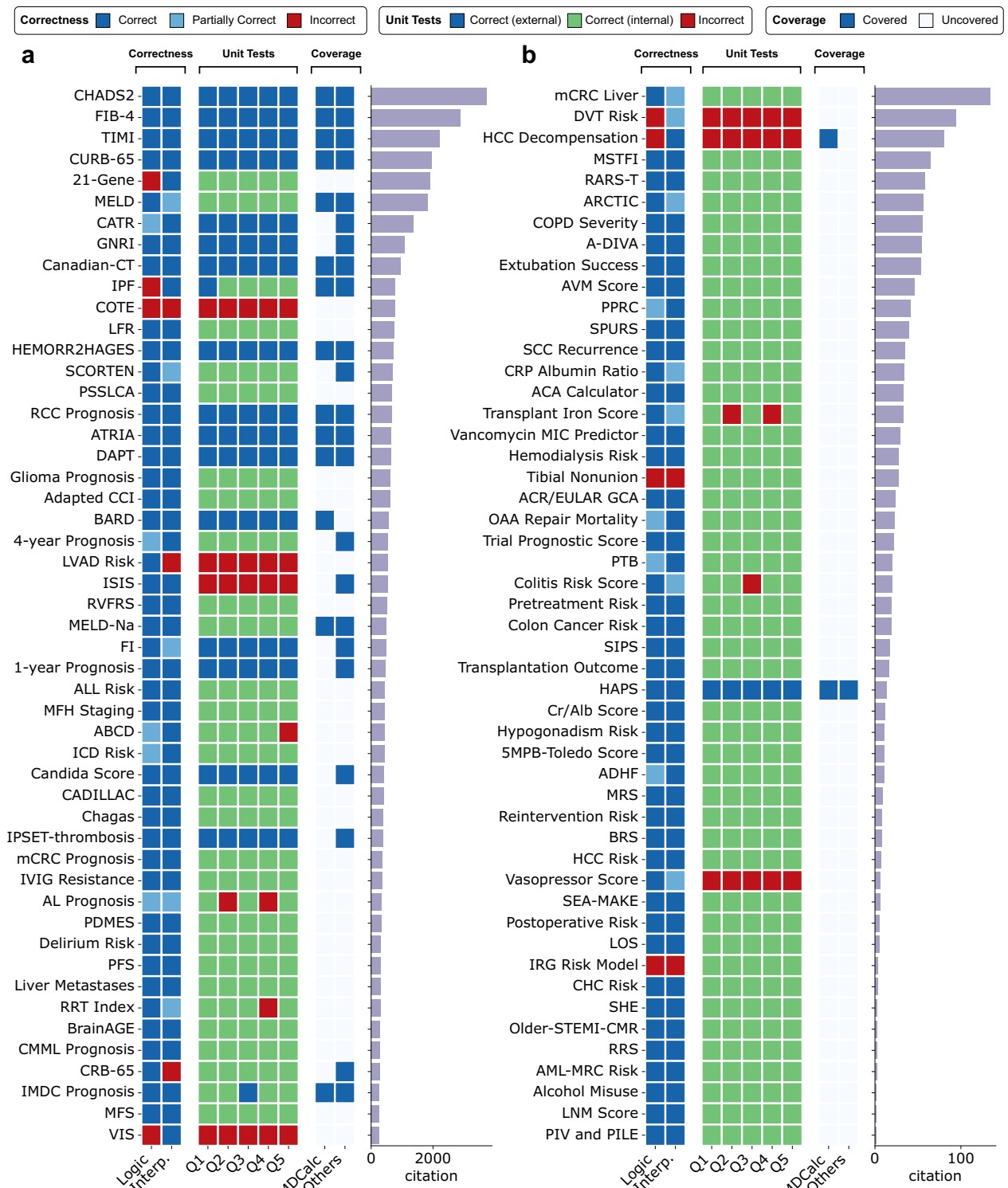

**Fig. 2 | Quality and coverage analysis of RiskCalcs. a** Evaluation results of the top-50 most cited calculators in RiskCalcs. **b** Evaluation results of a random sample of 50 calculators in RiskCalcs. Logic: computing logics; Interp.: result interpretation. Q1–Q5 denote unit test questions (clinical vignettes) for each tool. Source data are provided as a Source Data file.

AgentMD results by the 1039 risk calculators that have been applied to the patients (Fig. 5b). For each calculator, AgentMD ranks the eligible patients by a set of metrics concerning the specificity, severity, urgency, and absence from the note (more details can be found in the Methods section). Figure 5c shows the number of applied calculators per patient, which approximately follows a normal distribution with a mean value of 4.6. On the other hand, the number of eligible patients per tool follows a long-tail distribution (Fig. 5d), with less than 100 eligible patients for most calculators. These results show that multiple clinical calculators can be considered by AgentMD at the same time for

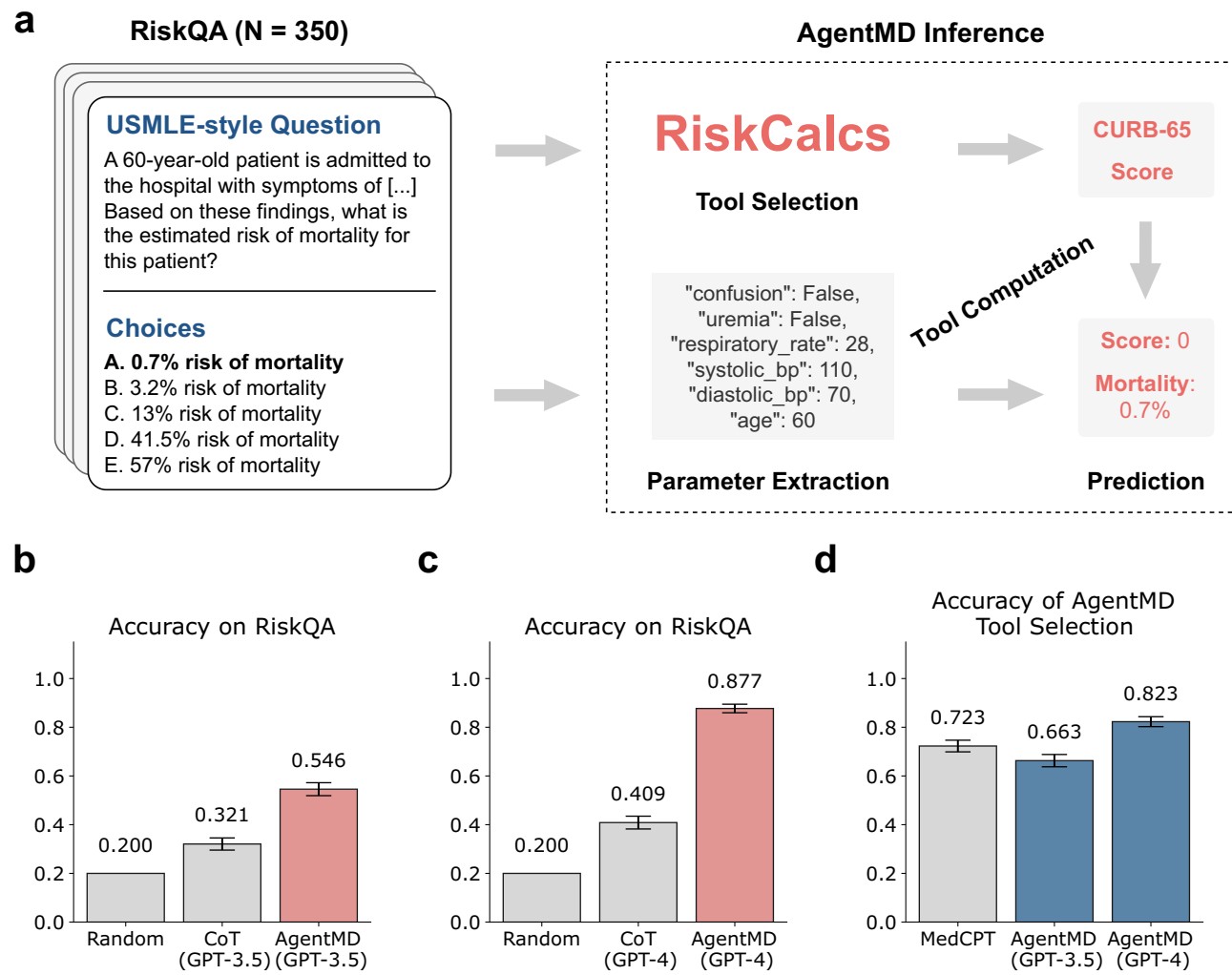

**Fig. 3 | Evaluations of AgentMD on RiskQA. a** Example of a question in RiskQA and how AgentMD answers it. **b** The performance of GPT-3.5-based AgentMD compared to Chain-of-Thought (CoT) prompting on RiskQA. **c** The performance of GPT-4-based AgentMD compared to CoT prompting on RiskQA. **d** The accuracy of tool selection by MedCPT and AgentMD. Accuracies (central lines) in (**b**–**d**) are defined as the proportions of correctly answered questions in all questions ($N = 350$). Error bars are the standard deviations of the accuracy proportions. Source data are provided as a Source Data file.

a patient, which provides more comprehensive risk assessments than the current stand-alone usage of clinical calculators.

Figure 5e illustrates the distribution of patient results for the two most commonly applied calculators by AgentMD. The first calculator predicts the short-term mortality of acute exacerbation of chronic heart failure[24]. While the mean specificity is low, which indicates that most of the needed parameters are missing from the patient notes, its urgency and severity distributions have higher mean values. The absence distribution of the calculator has two peaks – the higher one close to 100 and the lower one close to 0 – which indicates that short-term mortality is not assessed in most of the eligible patient notes. The second calculator predicts 4-year mortality in older adults[25]. Unlike the short-term mortality, most patient results for the 4-year mortality prediction are not urgent, and the severity also distributes differently, with a lower mean value. As expected, they are mostly absent from the patient notes. These two examples demonstrate how different calculator results can provide distinct insights regarding the specific risks of the eligible population.

We have also evaluated whether AgentMD computation results can improve in-hospital mortality prediction, which is an important and widely studied outcome in healthcare. Specifically, we compare AgentMD to the vanilla GPT-4 as recent studies[26–29] have demonstrated its capabilities in predicting clinical risks. Both AgentMD and GPT-4 perform zero-shot predictions in this setting. For the patient cohorts corresponding to each calculator, we draw the receiver operating characteristic (ROC) curves and compute the areas under the ROC (AUC) for predicting in-hospital mortality. Among the 1039 used ones, 604 calculators have sub-cohorts with at least one observed in-hospital death. Notably, our screening of these calculators discovered 113 useful clinical calculators curated and used by AgentMD that have higher AUC than the vanilla GPT-4. Figure 5f shows a sample of four such tools, which cover various scenarios such as high-risk varices and non-ST-elevation myocardial infarction (NSTEMI). Our results indicate that the in-hospital mortality prediction might improve with the AgentMD computation results for the patients who are eligible for these calculators.

## Discussion

In this study, we address two critical issues in clinical tool learning with LLMs: the absence of a comprehensive toolbox, alongside a deficiency in methodologies and evaluations for tool application. Motivated by the fact that many clinical calculators are reported in the biomedical literature, we used PubMed as the knowledge source to curate the

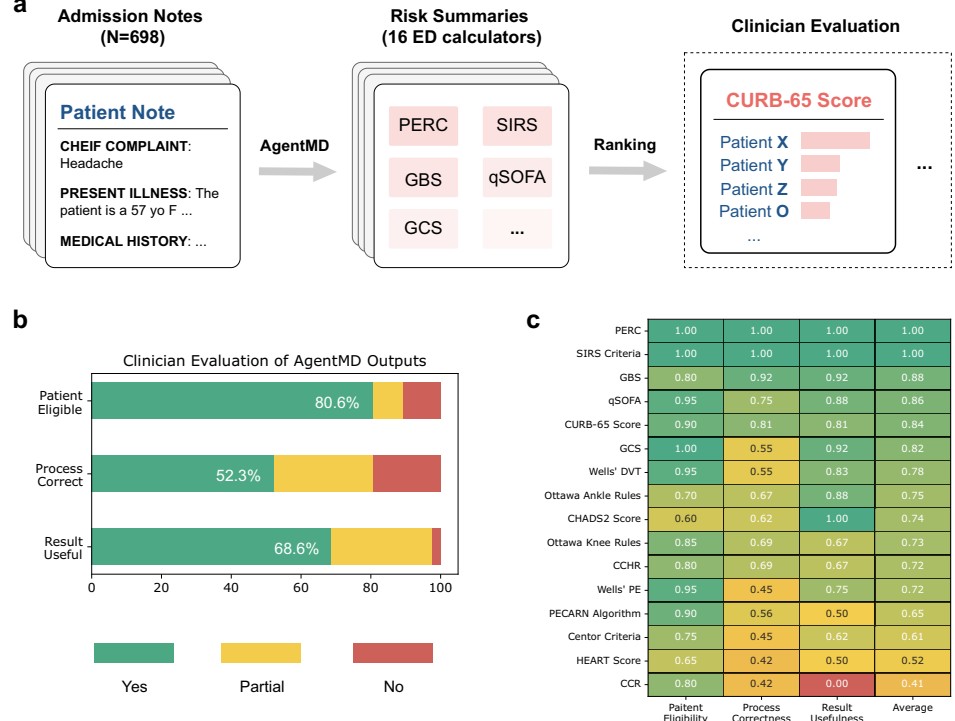

**Fig. 4 | Individual-level evaluation results on the emergency department provider notes. a** AgentMD is applied to emergency department provider notes from Yale Medicine with a toolkit of 16 commonly used calculators. For each calculator, the patients are then ranked by the overall risk, and the top 5 patients are selected for evaluation. **b** The distributions of clinician annotations on all patient-calculator pairs. **c** The per-calculator evaluation results averaged over patients and annotators, ranked by the average score. Source data are provided as a Source Data file.

clinical tools at scale. Exploiting the language and code generation capabilities of LLMs, AgentMD first curates over 2000 tools in Risk-Calcs using PubMed abstracts. Users can assess the quality of each curated tool using the associated publication, the population size, and the extracted utility metrics. Our manual evaluations on a representative subset of 100 calculators have demonstrated the high quality of RiskCalcs, scoring over 85% on three quality metrics and passing 91.6% of the unit tests. The most common cause of an incorrect or partially correct computing logic and result interpretation is the lack of executable Python functions and certain result interpretations, respectively.

RiskCalcs also covers various clinical tools that have not been implemented elsewhere on the Web, showing its potential as a supplement to the existing hubs. The clinical calculation tools are implemented as reusable programs by AgentMD, which results in both generalizability for different LLMs and sufficient precision for computation. Concurrent efforts such as OpenMedCalc[30] adopt manual curation and restrict the implementation schema, which is not scalable and might not be applicable to other LLMs than GPTs. On the other hand, Almanac[31] implements the use of certain calculators as retrieval-augmented generation with their raw textual descriptions, which can be imprecise due to the lack of programmatic executions of arithmetic.

Experimental results on the RiskQA benchmark show that AgentMD can accurately select and use clinical calculators, outperforming GPT-4 by a large margin. Specifically, our experimental results on RiskQA demonstrate that while GPT-4-based AgentMD achieved the tool selection accuracy of 0.823, the performance decreases to 0.663 with GPT-3.5 as the backbone LLM, indicating room for further improvements. In addition, evaluations on the ED provider notes show that AgentMD can accurately

perform risk calculation at the individual level, with slightly lower performance than the well-controlled RiskQA, mainly due to the missing value issues with the real-world data. Finally, the application of AgentMD on large-scale admission notes from MIMIC-III can provide unique insights for hospital-level risk management. As shown in Supplementary Table 13, many clinical calculators require non-standardized input parameters that can only be extracted from the patient notes, which necessitates the use of language agents for automatically using clinical calculators.

Although our research highlights the potential of clinical language agents like AgentMD, it is subject to several limitations. First, the creation of calculator tools was restricted to PubMed abstracts, overlooking detailed descriptions in full-text articles due to accessibility issues. Future work should aim to broaden the data sources for tool development. Second, the utilization of GPT-4 as the core LLM in AgentMD introduces a notable limitation due to its high operational costs and the challenge of deploying it locally. This constraint underscores the potential benefits of investigating alternative, possibly open-source LLMs like Llama[32], which may offer more cost-effective and flexible deployment options while maintaining adherence to stringent data protection standards. In addition, our work focuses on risk predictions from the text modality, and incorporating other modalities such as structured data and images remains an important future direction for exploration. The Supplementary Materials contain a brief analysis of the impact of structured data[33]. While AgentMD has been evaluated for both tool curation and tool usage capabilities with various datasets, it should be noted that further and more comprehensive evaluations are required before it can be incorporated into the clinical workflow.

In conclusion, AgentMD represents a promising methodical approach to enhancing clinical decision-making through the

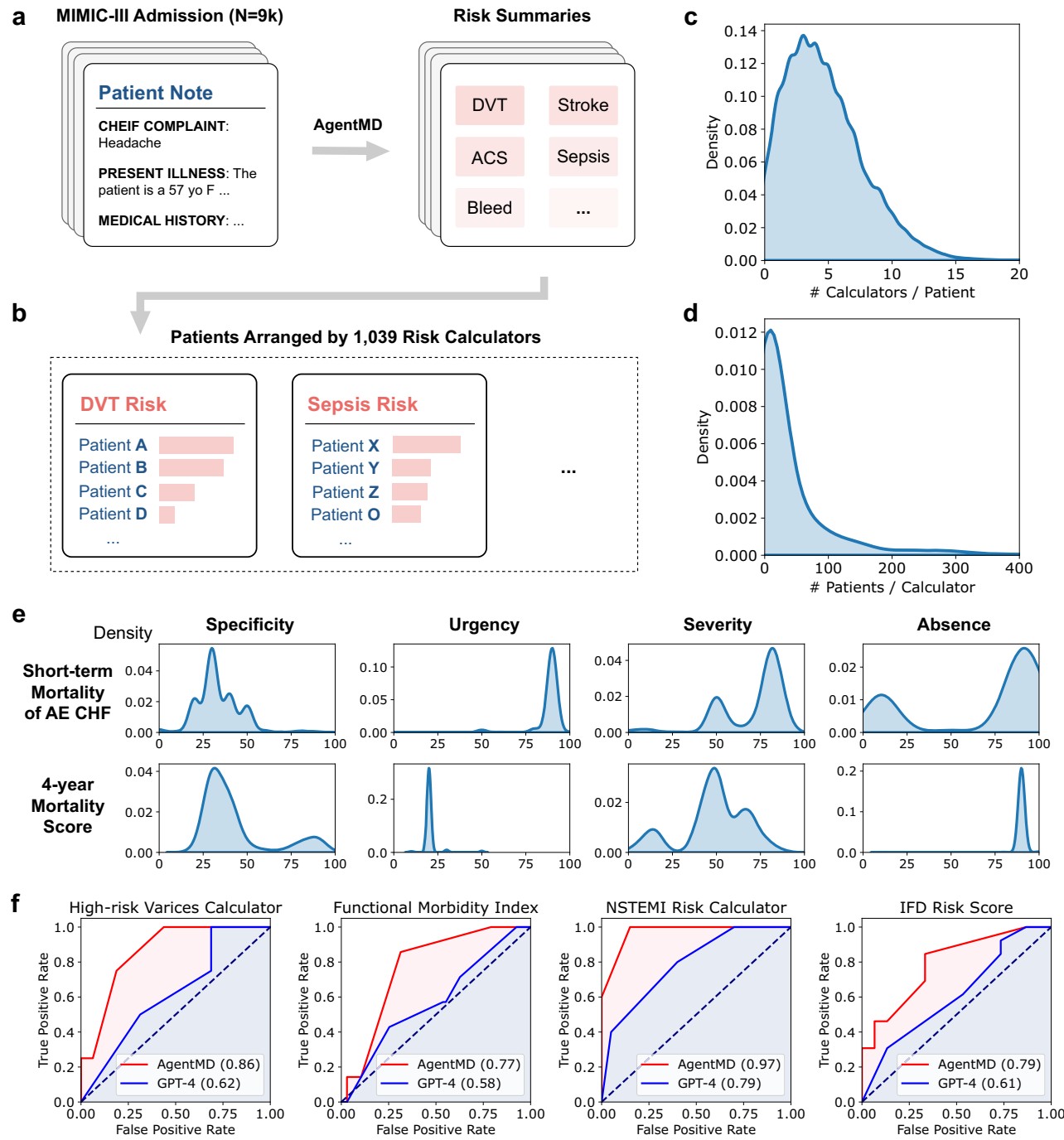

**Fig. 5 | Applying AgentMD on the MIMIC-III cohort. a** AgentMD is applied to 9822 admission notes in MIMIC. **b** AgentMD calculation results are aggregated by the risk calculators, and patients are ranked within each tool. **c** The distribution of the number of selected calculators for each patient. **d** The distribution of the number of eligible patients for each used calculator. **e** Calculation result distributions from two calculators. **f** Sample ROC curves of clinical calculators where AgentMD outperforms GPT-4 in predicting in-hospital mortality. Source data are provided as a Source Data file.

automated generation and application of a comprehensive set of clinical calculators, RiskCalcs, derived from PubMed articles.

## Methods

### Compliance statement

Our research complies with all relevant ethical regulations. The experiments were performed on computation platforms compliant to Health Insurance Portability and Accountability Act and approved by the Yale institutional review board (IRB) with the protocol

#2000035077, with a waiver of informed consent due to the retrospective use of de-identified patient data.

### Backbone LLMs

In this study, we use GPT-3.5 (1) (model index: gpt-35-turbo) and GPT-4 (2) (model index: gpt-4) from OpenAI as the backbone LLMs for AgentMD and compared baselines, because they have been shown to achieve state-of-the-art performance in both general and biomedical domain (3). We access these models

through the Application Programming Interface (API) of Azure OpenAI services, which is compliant with the Health Insurance Portability and Accountability Act (HIPAA). We use the model version 0613 for both GPT-3.5 and GPT-4, and the API version of 2023-07-01-preview. We set the decoding temperature to be 0.0 across this study to ensure deterministic outputs.

## Curating large-scale medical calculators in AgentMD

AgentMD uses three steps to automatically curate risk calculator tools from PubMed: article screening, calculator drafting, and calculator verification. One example of the AgentMD-generated calculator in RiskCalcs is shown in Fig. 1a.

During article screening, AgentMD first applies a Boolean query (patient AND (risk OR mortality) AND (score OR point OR rule OR calculator)) to search for recent articles between January 2000 and April 2023 in PubMed. This search resulted in a total of 339,952 articles. AgentMD uses GPT-3.5-Turbo, an efficient and capable LLM, to screen them and finds 33,033 articles that potentially describe a new risk score or calculator. The prompt for screening PubMed abstracts is shown below:

User:

Here is a PubMed article:

{PubMed article title}

{PubMed article abstract}

Does this article describe a new risk score or risk calculator? In healthcare, a risk score quantitatively estimates the probability of a clinical event or outcome, such as disease development or progression, within a specified period. These scores are derived from algorithms using variables like patient demographics, clinical history, laboratory results, and other relevant health indicators. They aid clinicians in decision-making, allowing for personalized patient care and resource allocation. Simply answer with "yes" or "no":

GPT-3.5-Turbo

{Output}

Then, AgentMD uses GPT-4, the state-of-the-art LLM at the time of this work, to draft 24,190 structured medical calculators from these articles, where each calculator includes key sections such as title, purpose, patient eligibility, medical topics, computing logics (implemented as functions in Python), interpretation of results, and clinical utility. The prompt for drafting the calculator is shown below:

System

You are a helpful assistant programmer for medical calculators. Your task is to read a PubMed article about a medical calculator, and if applicable, write a two-step calculator: (1) calculate a risk score based on multiple criteria; (2) interpret different ranges of the computed risk score into probabilities of risks.

User

Here is a PubMed article:

{PubMed article title}

{PubMed article abstract}

Does the article describe a simple two-step risk calculator, where the first step is to compute a risk score, and the second step is to interpret different risk scores? If no, please directly and only output "NO". Otherwise, please standardize the calculator into:

#Title

The name of the calculator(s).

##Purpose

Describe when this calculator should be used.

##Specialty

should be a list of calculator types, one or more of {specialty list}, separated by ",".

##Eligibility

Describe what patients are eligible.

##Size

The exact number of patients used to derive this calculator. Only put a number here without any other texts.

##Computation

Detailed instructions of how to use the calculator, including Python functions with clear docstring documentation. Please be self-contained and detailed. For example, if the computation involves multiple items, please clearly list each item. If one item has multiple possible values (e.g., 0–2), you also need to clearly define what each value means.

##Interpretation \n Should be a list, where each item describes the interpretation (actual risk) for a value or a range of the computed risk score.

##Utility

Evaluation results of the clinical utility of the risk score, such as AUC, F-score, PPV.

##Example

Generate a sample patient note and a detailed demonstration of using the calculator, and interpret the results. Think step-by-step here.

Please be as detailed as possible.

GPT-4

{Output}

Since the drafted calculators might contain inaccuracies such as hallucinations, AgentMD further uses GPT-4 to self-verify the generated calculators based on a list of six criteria shown below, such as consistency between the original evidence and the curated tool. Only drafted tools that pass all six criteria are included, resulting in a final set of 2164 calculators in the RiskCalcs collection. The prompts for verification of the drafted tools are shown below:

System

You are a critical evaluator for a calculator that's supposed to describe a PubMed article. The calculator might contain errors. Always respond in a JSON dict formatted as Dict{"reasoning": Str(critical_reasoning), "answer": Str(yes/no)}.

User

Here is the original PubMed article:

{PubMed article title}

{PubMed article abstract}

Here is the calculator that's supposed to describe the article above:

{Generated calculator}

**Prompt 1.** Are the parameters clearly defined in the #Computation? If a parameter can have different scores, the definitions for each score must be provided.

**Prompt 2.** Are the parameters defined exactly the same in the article and the calculator?

**Prompt 3.** Is the #Computation logic in the calculator fully based on the original article without any assumptions? Answer no if the article does not provide clear computing logics or weights.

**Prompt 4.** Is the #Interpretation of the calculator fully based on the original article without any assumptions? Score ranges and corresponding risks should be exactly the same between the calculator and the article.

**Prompt 5.** Is the #Interpretation of the calculator useful? A useful calculator should contain quantitative risk rates or qualitative risk groups for different score ranges.

**Prompt 6.** Is the calculator free from any bugs or other issue?

GPT-4

{Output}

## Evaluation of AgentMD as a tool maker

For tool making, we evaluated the correctness of quality checks and the pass rate for unit tests of RiskCalcs, as well as its coverage. Specifically, we selected the top 50 most cited calculators and a randomly sampled subset of 50 calculators from the rest of RiskCalcs for manual quality evaluation. The article citation counts were derived from the iCite API on December 17, 2023. Three annotators were recruited to annotate the correctness of the generated computing logic and the result interpretation with the labels of Correct, Partially Correct, and Incorrect, as well as coverage metrics (whether the calculator has been implemented as an online tools, with the labels of Covered or Not covered). The three annotators conducted the initial annotations independently. Then, they had a round of discussion for the disagreed annotations and determined consensus annotations. For unit tests, one annotator further conducted the computation of five synthetic parameter sets per validated calculator and compared the manually calculated results with the AgentMD computation results. If the AgentMD computation results are the same as the results from an online implementation (e.g., from an MDCalcs calculator), the label is Correct (external). If the results come from the annotator's computing based on PubMed abstracts, the label is Correct (internal). Otherwise, the label is Incorrect.

## Applying Medical Calculators to Patients with AgentMD

Figure 1b shows a high-level overview of how AgentMD computes risks from candidate calculators for any given patient in three main steps: tool selection, tool computation, and results summarization. In the tool selection step, AgentMD first retrieves the top ten most relevant calculators based on the matching between patient information and the calculator description with MedCPT[34], a foundation model for biomedical information retrieval. Then, AgentMD selects the eligible tools from the retrieved candidates using LLMs, with the goal of providing more accurate and explainable tool selections. The tool selection process is essential, as the interpretations of clinical calculators are based on the populations that participated in their original clinical studies. As such, calculators should only be applied to patients who meet the eligibility criteria. For each selected tool, AgentMD computes the risk of the patient by generating Python scripts via the corresponding calculator. During the tool computation process, AgentMD is provided with a Python interpreter to interact with – the environment returns the printed results or error messages when executing the code that AgentMD writes to use the clinical tools. Based on the returned information from the code interpreter, AgentMD either re-tries other code or summarizes the whole interaction history into a paragraph of the risk calculation results. If the required parameters from the clinical calculator are missing from the patient notes, AgentMD will make a range estimation based on the best- and worst-case scenarios.

## Chain-of-thought baseline

We have compared AgentMD to the Chain-of-Thought (CoT)[22] baseline in the evaluation with RiskQA. Specifically, CoT is a commonly used prompt engineering method that instructs the LLM to think step-by-step to solve the task. During the evaluation, AgentMD and the CoT baseline are provided with the same question inputs in RiskQA. Their only difference is that AgentMD can use the RiskCalcs toolkit, while the CoT baseline relies on the encoded knowledge within the backbone LLMs. The CoT prompt is shown below:

> System
>
> You are a helpful assistant. Your task is to answer a medical examination question. Please indicate your answer choice (A/B/C/D/E) at the end of your answer by "Therefore, the answer is A/B/C/D/E."
>
> User
> {RiskQA Question}
> {RiskQA Options}
> Let's think step-by-step.

## Evaluation of AgentMD as a tool user

Since there have not been any prior efforts in automating the selection and application of large-scale clinical calculators, we first manually create a new benchmark called RiskQA to evaluate the tool selection and usage capabilities of such systems. To construct the RiskQA dataset, we reused the 350 manually validated sets of parameters for unit tests of RiskCalcs calculators with correct computing logics and result interpretations. For each parameter set and validated calculation, we further used GPT-4 to expand them into a clinical vignette, possible choices, and the correct answer in the style of a multi-choice United States Medical License Examination question (one example is shown in Fig. 3a). The task of RiskQA is to select the correct choice given the patient description. Since the required calculator is not revealed in the question, RiskQA can also test the capability to select a suitable tool.

In addition to the well-controlled RiskQA dataset, we have also applied AgentMD to 698 provider notes from the emergency department of Yale Medicine to evaluate its accuracy on real patient data at the individual level. Specifically, we consider 16 commonly used calculators in the emergency department, and the calculator selection is implemented as a single in-context inference with all the calculator information. Then, AgentMD will apply the selected calculators to a given patient and rank the patients based on their final numeric outputs for each calculator. The results of the top five patients for each calculator are evaluated. Unlike the end-to-end evaluation adopted in RiskQA, three aspects for each patient-calculator pair are manually evaluated within this cohort: (1) whether the patient is eligible for using the calculator; (2) whether the calculation process is correct; and (3) whether the calculation result is helpful in the clinical setting. Each aspect is annotated with yes, partial, and no. Similar to the quality evaluation, each aspect can be annotated with three labels: yes, partial, and no. The experiment was performed on computation platforms compliant to Health Insurance Portability and Accountability Act and approved by the Yale institutional review board (IRB) with the protocol #2000035077.

Finally, we scale up to the entire set of 2,164 calculators in RiskCalcs and demonstrate the utility of AgentMD for population-level data analytics. Specifically, we use AgentMD on 9822 admission notes in MIMIC-III[19,35], a database of critical care electronic medical records. Given a patient note, AgentMD first generates a list of clinical risks that the patient might have. For each generated clinical risk, AgentMD retrieves relevant medical calculators from RiskCalcs and judges if the patient is eligible to use them. Finally, AgentMD applies the calculators to eligible patients and ranks them based on four LLM-predicted scores. These scores include: Specificity (0–100) denotes the confidence of the calculator result. Specificity is low if there are missing values and the range is wide between the risk scores of the best-case and worst-case scenarios. Specificity is high if there is no range estimation (best and worse case scenarios) and the risk calculator result contains only one specific score (not a range); Urgency (0–100) denotes whether the risk considered by the calculator is acute or chronic. Urgency is high if there is immediate danger worth medical attention. On the other hand, urgency is low if the risk is about 1-year or 5-year; Severity (0–100) denotes the extent of the calculated risk. Severity is high if the predicted risk probability is close to 100%. Severity is low if the predicted risk probability is close to 0%; Absence (0–100) denotes whether the calculated risk is missing in the original patient note. Absence is 100 if the calculator result (the risk) is not considered or reflected in the patient note. Absence is 0 if the calculator result has already happened or been considered in the patient note. To validate such scores generated by AgentMD, we have randomly sampled

30 pairs of computed patient risks. Each pair consists of the same calculator and two different patients. Two physicians were employed to annotate which patient has higher scores in each of the defined axis, with moderate levels of agreements: 86.7% (95% confidence interval (CI): 78.3–93.3%) for specificity, 61.7% (95% CI: 50.0–73.3%) for urgency, 76.7% (95% CI: 65.0–86.7%) for severity, and 66.7% (95% CI: 58.3–75.0%) for absence. On average, their annotations also align with those of AgentMD score predictions: 75.0% (95% CI: 66.7–82.5%) for specificity, 67.5% (95% CI: 59.2–75.8%) for urgency, 68.3% (95% CI: 60.0–76.7%) for severity, and 78.3% (95% CI: 70.8–85.0%) for absence. Confidence intervals are computed via bootstrapping with 10,000 times of resampling. These results show that the LLM-generated scores are reasonably aligned with physician judgment and can provide a meaningful and quantitative way of assessing clinical risks. Then, we conduct a screening study to check if any calculators can be potentially useful for in-hospital mortality prediction, given the patient admission information. Specifically, we use the sum of the specificity, urgency, and severity scores to represent the overall risk level for each patient-calculator pair. Overall, 1039 clinical calculators have been applied to at least one patient in the MIMIC dataset. Among these calculator cohorts, 604 have both positive and negative labels for in-hospital mortality, and we screened their corresponding tools for the in-hospital mortality prediction capability. Within the cohort eligible for each calculator, patients are ranked by their overall risk, and AUROC is used to evaluate the performance. Each tool is compared to the GPT-4 baseline which directly predicts patient mortality with chain-of-thought prompting, as prior studies have shown its effectiveness for this task[26–29].

## Statistics & reproducibility

No statistical method was used to predetermine sample size. No data were excluded from the analyses. The experiments were not randomized. The Investigators were not blinded to allocation during experiments and outcome assessment.

## Data availability

PubMed abstracts can be downloaded at https://ftp.ncbi.nlm.nih.gov/pubmed/baseline/. The MIMIC-III dataset is available at https://physionet.org/content/mimiciii/1.4/. The raw emergency department provider notes contain sensitive health information and cannot be publicly shared due to HIPAA and institutional policies. Controlled access may be granted to qualified researchers who enter into a data use agreement that prohibits re-identification or redistribution of the data and limits use to research purposes only. Requests should be directed to the corresponding author and will be addressed promptly. The RiskCalcs collection and the RiskQA dataset generated in this study have been deposited at: https://github.com/ncbi-nlp/Clinical-Tool-Learning. Source data are provided in this paper.

## Code availability

The MIMIC-III preprocessing code is available at https://github.com/bvanaken/clinical-outcome-prediction and https://github.com/YerevaNN/mimic3-benchmarks. The MedCPT encoders are available at https://github.com/ncbi/MedCPT. The Faiss package is available at https://github.com/facebookresearch/faiss. The source code of AgentMD is publicly available at: https://github.com/ncbi-nlp/Clinical-Tool-Learning[36].

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

## Acknowledgements

This research was supported by the Intramural Research Program of the National Institutes of Health (NIH). The contributions of the NIH author(s) are considered Works of the United States Government. The findings and conclusions presented in this paper are those of the author(s) and do not necessarily reflect the views of the NIH or the U.S. Department of Health and Human Services. This research was also supported by R01LM014604 (Q.C.) and 1K99LM014903 (Q.J.). We thank Charalampos Floudas and Balu Bhasuran for helpful discussions.

## Author contributions

Q.J., Q.C. and Z.L. designed the study. Q.J. conducted the data collection, model construction, model evaluation, and manuscript drafting. Z.W., Y.Y., N.K., N.W., X.A. and Z.H. carried out the data collection and analysis. Z.W., Y.Y., Q.Z., R.T., D.W., T.H. and W.J.W. contributed to the data annotation. Z.L. supervised the study. All authors contributed to writing the manuscript and approved the submitted version.

## Competing interests

The authors declare no competing interests.
