## [Transparent Peer Review file · Nature Communications]

AgentMD: Empowering Language Agents for Risk Prediction with Large-Scale Clinical Tool Learning

Corresponding Author: Dr Zhiyong Lu

Version 0:

Reviewer comments:

Reviewer #1

(Remarks to the Author)

The paper proposes an LLM-based AI agent that (1) automatically creates clinical calculators from public research articles, (2) selects and applies them based on individual patient information and summarizes the results. The quality of the created toolset is verified through unit tests on selected samples. The effectiveness of tool usage is demonstrated with both manually crafted test data and real test cases in an end-to-end manner.

The paper is well-written, with solid evaluations and clear presentation. The idea of applying an LLM-based agent for tool creation is novel. Overall, I think the paper showcases a valuable application of an LLM-based autonomous agent in scientific research, with significant benefits for risk prediction.

The methodology appears sound, and I did not detect any clear flaws in the data analysis or evaluation.

Some issues in the paper, if addressed, could further enhance its quality:

1. Tool Correctness Testing: For evaluating tool correctness, the unit test cases are automatically generated by GPT-4. Since ensuring tool accuracy is critical, it would be beneficial to include a small set of manually crafted, high-quality test cases created by human experts for unit testing. Testing with extreme cases and boundaries would also strengthen the evaluation.
2. Tool Creation Framework Design: The design of the tool creation agent framework is somewhat unclear. Specifically, it is not clear how the agent can reliably create calculation functions by referencing only the abstracts of the collected papers. I would expect the calculation details to be found primarily in the main body of the papers. Providing more details on this in the Methods section would help clarify the framework.
3. Baseline Methodology: The baseline method, COT with GPT-4, is not clearly explained. Are the same tools used for the baseline, and how is prompting within COT? Providing more detail on the baseline setup and the prompting used in COT would clarify this comparison.

Additionally, I feel that one potential risk of the proposed method is the use of proprietary models for diagnostics. Sending patient information to the API may lead to a potential leak of private information and raise privacy concerns. This potential risk could be discussed in the paper.

Despite the aforementioned potential issues, I believe the paper is of high quality and is overall valuable to the field.

(Remarks on code availability)

--Both the created toolset (RiskCalcs) and the evaluation dataset (RiskQA) are provided in the codebase, with clear instructions on how to perform the agent's evaluation.

--The toolset creation pipeline does not appear to be open-sourced in the code.

Reviewer #2

(Remarks to the Author)

This manuscript describes an agent-based large language model (LLM) for building and applying clinical risk calculators to

clinical notes.

The authors use automated scripts to screen through 37M PubMed articles for candidate articles that describe a clinical risk calculator.

The candidates are further narrowed down to 2,164 using a combination of GPT-3.5-Turbo and GPT-4.

Next, LLMs are used to convert the description of these tools to python codes for each clinical risk calculator.

Given a patient note, AgentMD model (based on ChatGPT) selects one or more risk calculator to apply.

They perform qualitative assessment of the usefulness or appropriateness of the risk predictions, using three datasets namely 1) a manually create a new benchmark called RiskQA, 2) emergency provider notes from Yale Medicine, and 3) sample admission notes from the MIMIC-III database. Overall the proposed approach is innovative and AgentMD introduces a new paradigm to search biomedical literature/clinical text for useful clinical risk calculator/models.

1. While the proposed approach has merits as a data mining tool, it's far from being a clinically useful tool and this point needs to be clearly emphasized in the manuscript.
2. A key concern is that the authors don't make any attempt to assess AgentMD's level of randomness in risk calculator selection, signs/symptoms extraction, and risk calculations.
3. In the introduction they discuss "evidence-based risk assessment" but it's not clear what the level of evidence for each risk calculator is (although they consider the number of citations as a proxy), and whether the proposed risk calculators were validated across different patient cohorts, prospectively tested, and have shown clinical utility.
4. Another key consideration is the mapping of risk score input data to standardized codes with appropriate units of measurement, etc. The generated python code does not seem to take measurement units as well as other factors such as the precise method of measurement or site of measurement into account (e.g., blood pressure was measured from from what body site, what method, in what position?).
5. The RiskQA dataset seems like a good resource, however, it's not clear how the correct risk level was assigned to each scenario.
6. The MIMIC-III evaluation could benefit from comparing the predicted risk against the ground truth for the top 5 conditions (or even a randomly selected set of easily obtainable outcomes). For instance, comparing the predicted risk of hospital mortality against the observed mortality and reporting AUC.

(Remarks on code availability)

I reviewed the LLM prompts in the appendix and they look reasonable.

Reviewer #3

(Remarks to the Author)

The authors present AgentMD: an LLM-powered system to (1) curate clinical calculators from PubMed, (2) perform patient risk assessment through agent-tool use, and (3) provide population-level risk insights. The article is well written and the curation of clinical calculators at scale is novel and compelling. There are several issues, particularly in the evaluation of patient and population risk assessment (2 and 3 above), that should be addressed prior to acceptance into a high-impact journal such as Nature Communications:

1. It is not clear to me why the generated calculators are only assessed on unstructured notes. I would imagine the immediate utility of these calculators would be their application to structured EHR data. For example, leveraging all structured data within the first 3 hours of presentation to the ED of Yale Medicine as opposed to only the ED provider notes.
2. I don't believe a reasonable baseline is provided to assess the benefit of tool use by the LLM. Would running all calculators (computationally cheap) and then selecting the top-5 patients by overall risk result in better or worse performance than what is presented?
3. It would be beneficial if the authors could provide more objective measures of risk prediction performance.
4. I don't believe the Specificity, Urgency, Severity, and Absence categories generated by the LLM are properly evaluated. What is the evidence that suggests these are meaningful for population-level risk insight?

(Remarks on code availability)

The repository is well-organized and the code is well-structured and readable. I commend the authors on providing extensive source code.

Version 1:

Reviewer comments:

Reviewer #1

(Remarks to the Author)

I've reviewed the authors' responses to my previous comments and appreciate their efforts in incorporating manually crafted test cases by domain experts, as well as providing more details about the toolset creation pipeline.

Overall, I maintain my previous positive assessment of this paper. I believe the proposed AI agent-based clinical tool represents a valuable contribution to the field, and the revised manuscript shows noticeable improvement. While I still think that relying solely on collected paper abstracts for function creation limits the method, I acknowledge that this is an area that can be improved in future work.

(Remarks on code availability)

1. Both the created toolset (RiskCalcs) and the evaluation dataset (RiskQA) are provided in the codebase, with clear instructions on how to perform the agent's evaluation.
2. According to the author, additional details have been added regarding the toolset creation pipeline.

Reviewer #2

(Remarks to the Author)

Thank you for your detailed response and additional data.

Overall this is an interesting paper. As I mentioned in my previous review, this work presents a novel application of LLMs and agents for data mining of published literature. However, it's important not to overstate the clinical utility of the proposed approach.

The evaluation methods remain weak, and include certain amount of hand-picking. It's not clear to me if RiskQA is a valid dataset of risks. ChatGPT is not an appropriate baseline for mortality prediction. There are a number of other methods based on structured data that can present better baselines. <https://github.com/YerevaNN/mimic3-benchmarks>

(Remarks on code availability)

Codes are appropriate.

Reviewer #3

(Remarks to the Author)

I thank the authors for their thoughtful responses to my comments. However, I don't believe the revisions have sufficiently addressed the issues identified in the previous round of review:

Major Comments:

1. The addition of Supplementary Table 13 supports the utility of unstructured input extraction but does not sufficiently support the omission of structured inputs. While the authors present that 68.75% of clinical calculators require at least one unstructured input parameter, they do not explore rates of data missingness. I would expect that a substantial percentage of non-missing inputs to the clinical calculators would be present in structured data but not the clinical notes. It would be helpful if the authors could provide this analysis to support their approach. Further, the highlighted challenges around non-standardized input parameters and poor integration with electronic health records, while important, are orthogonal to the omission of structured data. External labs, vital-signs, bedside monitors, etc. enter the EHR as structured data. These values are often imported into clinical notes via dotphrases, etc.
2. I thank the authors for correcting my assumption that running all calculators would be computationally trivial. I would also like to clarify my previous comment regarding a reasonable baseline to assess the benefit of tool-use by the LLM. The RiskQA experiment is cogent and CoT as a baseline supports the claim that AgentMD's tool-use improves performance on USMLE-type questions. My concern centers around the claim that "AgentMD accurately computes medical risks with real-world patient data at an individual level." Here, no baseline is provided to assess whether "AgentMD has the potential to assist the risk assessment process by automatically selecting and applying medical calculators to inform the risks." To validate that the process of tool selection by AgentMD improves risk assessment, a reasonable baseline might include running all 16 calculators and presenting the top 5 patients by overall risk. I would expect this to be computationally feasible given the larger experiment on population-level risk insights.
3. I commend the authors on their inclusion of the in-hospital mortality prediction task which greatly strengthens the manuscript. The experiment could benefit from additional clarity in the Methods section. My understanding of the experiment, as written, is that 604 overlapping cohorts are evaluated corresponding to patients that were assigned one of 1039 calculators by AgentMD and had at least one in-hospital mortality in the cohort. Of these, AgentMD outperformed GPT-4 in 114 cohorts. It would be helpful to evaluate performance across the entire population (with the AgentMD risk score being potentially 0 for some patients).
4. I again commend the authors on their validation of the Specificity, Urgency, Severity, and Absence categories generated by the LLM against physician judgement. To support the claim that "these results show that the LLM-generated scores are reasonably aligned with physician judgment and can provide a meaningful and quantitative way of assessing clinical risks," confidence intervals should be provided so the reader can assess the significance relative to random chance.

Minor Comments

1. The addition of the 100 manually curated unit tests strengthens the manuscript. Some of the details listed in the Results section should be in the Methods instead.

(Remarks on code availability)

The repository is well-organized and the code is well-structured and readable. I commend the authors on providing extensive source code.

Version 2:

Reviewer comments:

Reviewer #2

(Remarks to the Author)

Thank you for additional additional results with structured data. I have no further questions or comments.

(Remarks on code availability)

Reviewer #3

(Remarks to the Author)

I thank the authors for their thoughtful response to my comments. This will be a meaningful contribution to the field. My only remaining minor comment is that the physician agreement of urgency of MIMIC-III patients appears to not be significantly different than random chance (95% CI: 50.0–73.3%). I would not consider this “moderate to high levels of agreement.”

(Remarks on code availability)

The repository is well-organized and the code is well-structured and readable. I commend the authors on providing extensive source code.

DEPARTMENT OF HEALTH & HUMAN SERVICES

Public Health Service

National Institutes of Health
National Library of Medicine
Bethesda, Maryland 20894

March 30, 2025

Dear Reviewers,

We sincerely thank you for the thorough evaluation and insightful comments on our manuscript titled “AgentMD: Empowering Language Agents for Risk Prediction with Large-Scale Clinical Tool Learning” submitted to Nature Communications. We deeply appreciate the time and effort you invested in reviewing our work, as well as your positive remarks and constructive feedback. Your comments have been instrumental in improving the clarity, rigor, and overall quality of the manuscript. We have carefully and systematically addressed each comment and suggestion, and we believe the revisions have significantly strengthened our work. In the following sections, we provide a point-by-point response to each of the comments raised by the reviewers, along with the corresponding changes made to the manuscript.

Zhiyong Lu, Ph.D., FACMI, FIAHSI
Senior Investigator
National Library of Medicine (NLM)
National Institutes of Health (NIH)
8600 Rockville Pike
Bethesda, MD 20894 USA

Response to Reviewer #1

Overall Comment

The paper proposes an LLM-based AI agent that (1) automatically creates clinical calculators from public research articles, (2) selects and applies them based on individual patient information and summarizes the results. The quality of the created toolset is verified through unit tests on selected samples. The effectiveness of tool usage is demonstrated with both manually crafted test data and real test cases in an end-to-end manner.

The paper is well-written, with solid evaluations and clear presentation. The idea of applying an LLM-based agent for tool creation is novel. Overall, I think the paper showcases a valuable application of an LLM-based autonomous agent in scientific research, with significant benefits for risk prediction.

The methodology appears sound, and I did not detect any clear flaws in the data analysis or evaluation.

Response

We are grateful for your positive remarks on our manuscript. We have provided point-by-point responses to your concerns in the following sections.

Comment 1 (Tool Correctness Testing)

For evaluating tool correctness, the unit test cases are automatically generated by GPT-4. Since ensuring tool accuracy is critical, **it would be beneficial to include a small set of manually crafted, high-quality test cases created by human experts for unit testing.** Testing with extreme cases and boundaries would also strengthen the evaluation.

Response

Thank you for your suggestion. We agree that incorporating more challenging and realistic unit tests is valuable for evaluating the quality of the curated tools. In response to your comment, we randomly selected 10 clinical calculators and engaged two medical student annotators to manually curate 10 clinical vignettes for each, resulting in 100 additional unit tests. These test cases were specifically designed to represent challenging or edge cases, where patient parameters fall near clinical decision boundaries.

In the revision, we have added in the Results / RiskCalcs Tools Feature High Quality and Extensive Coverage section: “In addition to GPT-4-generated sets, 100 sets of unit test parameters have been manually curated for ten randomly sampled calculators. For this manually curated subset, we specifically included more challenging and edge cases where the patient parameters fall close to the decision boundaries. As a result, AgentMD achieves a passing rate of 84.0%, which is slightly lower but remains overall consistent with the passing rate on the automatically generated unit tests.”

Comment 2 (Tool Creation Framework Design)

The design of the tool creation agent framework is somewhat unclear. Specifically, it is not clear how the agent can reliably create calculation functions by referencing only the abstracts of the collected papers. I would expect the calculation details to be found primarily in the main body of the papers. **Providing more details on this in the Methods section would help clarify the framework.**

Response

Thank you for your thoughtful comment. We agree that calculation details are often more thoroughly described in the full-text versions of articles. However, many of these articles are behind publisher paywalls, limiting our ability to access them for automated processing. While we had previously noted this constraint, we have now explicitly emphasized it in the revised manuscript: “First, the creation of calculator tools was restricted to PubMed

abstracts, overlooking detailed descriptions in full-text articles due to accessibility issues. Future work should aim to broaden the data sources for tool development.”

Figure. An example abstract that contains the calculation and interpretation details of the CURB-65 score.

Multicenter Study > Thorax. 2003 May;58(5):377-82. doi: 10.1136/thorax.58.5.377.

Defining community acquired pneumonia severity on presentation to hospital: an international derivation and validation study

W S Lim¹, M M van der Eerden, R Laing, W G Boersma, N Karalus, G I Town, S A Lewis, J T Macfarlane

Affiliations + expand
PMID: 12728155 PMID: PMC1746657 DOI: 10.1136/thorax.58.5.377

Abstract

Background: In the assessment of severity in community acquired pneumonia (CAP), the modified British Thoracic Society (mBTS) rule identifies patients with severe pneumonia but not patients who might be suitable for home management. A multicentre study was conducted to derive and validate a practical severity assessment model for stratifying adults hospitalised with CAP into different management groups.

Methods: Data from three prospective studies of CAP conducted in the UK, New Zealand, and the Netherlands were combined. A derivation cohort comprising 80% of the data was used to develop the model. Prognostic variables were identified using multiple logistic regression with 30 day mortality as the outcome measure. The final model was tested against the validation cohort.

Results: 1068 patients were studied (mean age 64 years, 51.5% male, 30 day mortality 9%). Age ≥ 65 years (OR 3.5, 95% CI 1.6 to 8.0) and albumin < 30 g/dl (OR 4.7, 95% CI 2.5 to 8.7) were independently associated with mortality over and above the mBTS rule (OR 5.2, 95% CI 2.7 to 10). A six point score, one point for each of Confusion, Urea > 7 mmol/l, Respiratory rate ≥ 30 /min, low systolic (< 90 mm Hg) or diastolic (≤ 60 mm Hg) Blood pressure, age ≥ 65 years (CURB-65 score) based on information available at initial hospital assessment, enabled patients to be stratified according to increasing risk of mortality: score 0, 0.7%; score 1, 3.2%; score 2, 3%; score 3, 17%; score 4, 41.5% and score 5, 57%. The validation cohort confirmed a similar pattern.

Conclusions: A simple six point score based on confusion, urea, respiratory rate, blood pressure, and age can be used to stratify patients with CAP into different management groups.

Nonetheless, we found that a substantial number of PubMed abstracts contain sufficient information to curate medical calculators. For example, in our submission we include Figure 1 based on the abstract from “Defining community acquired pneumonia severity on presentation to hospital: an international derivation and validation study” (PMID:12728155), which clearly describes the CURB-65 scoring criteria and its interpretation. This example demonstrates that, with appropriate quality control, PubMed abstracts can serve as a viable and valuable source for tool curation.

In response to your suggestion, we have added the full prompt templates used for screening PubMed abstracts, drafting medical calculators, and verifying tool correctness to the Methods section. We believe these additions provide greater transparency and help clarify how AgentMD curates the RiskCalcs toolkit from PubMed abstracts.

Comment 3 (Baseline Methodology)

The baseline method, COT with GPT-4, is not clearly explained. Are the same tools used for the baseline, and how is prompting within COT? **Providing more detail on the baseline setup and the prompting used in COT would clarify this comparison.**

Response

Thank you for your question and suggestion. In our evaluation, both AgentMD and the chain-of-thought (CoT) baseline were provided with the same question inputs from the RiskQA benchmark. The key difference is that AgentMD has access to the RiskCalcs toolkit, enabling it to compute results using curated tools, whereas the CoT baseline relies solely on the internal reasoning capabilities of the backbone LLM (GPT-4), without external tool use.

To clarify this comparison, we have added a new Chain-of-thought Baseline subsection to the Methods section in the revised manuscript. Additionally, we now include the exact prompt used for the CoT baseline to ensure full transparency.

Comment 4

Additionally, I feel that one potential risk of the proposed method is the use of proprietary models for diagnostics. **Sending patient information to the API may lead to a potential leak of private information and raise privacy concerns. This potential risk could be discussed in the paper.**

Despite the aforementioned potential issues, I believe the paper is of high quality and is overall valuable to the field.

Response

Thank you for your positive remarks and thoughtful comments. We take patient privacy extremely seriously and have implemented the following safeguards to minimize any risk of data leakage:

1. We accessed the GPT models via the Microsoft Azure OpenAI Service, which is compliant with the Health Insurance Portability and Accountability Act (HIPAA). This ensures that all transmitted data is subject to strict security and privacy protections, including encryption in transit and data handling policies that align with HIPAA standards.
2. All real-world clinical notes from Yale Medicine were thoroughly de-identified prior to being sent to the API, removing any patient-identifiable information.

In the revised manuscript, we have included the following clarification in the Methods / Backbone LLMs section: “We access these models through the Application Programming Interface (API) of Azure OpenAI services, which is compliant with the Health Insurance Portability and Accountability Act (HIPAA).”

Remarks on code availability

--Both the created toolset (RiskCalcs) and the evaluation dataset (RiskQA) are provided in the codebase, with clear instructions on how to perform the agent's evaluation.

--The toolset creation pipeline does not appear to be open-sourced in the code.

Response

Thank you for your acknowledgement of our codebase that contains the toolkit and the evaluation dataset. In the revision, we have included all the prompts used in the toolset creation pipeline in the Methods section. We have also updated our code repository to include the tool curation pipeline scripts.

Responses to Reviewer #2

Overall Comment

This manuscript describes an agent-based large language model (LLM) for building and applying clinical risk calculators to clinical notes.

The authors use automated scripts to screen through 37M PubMed articles for candidate articles that describe a clinical risk calculator.

The candidates are further narrowed down to 2,164 using a combination of GPT-3.5-Turbo and GPT-4.

Next, LLMs are used to convert the description of these tools to python codes for each clinical risk calculator.

Given a patient note, AgentMD model (based on ChatGPT) selects one or more risk calculator to apply.

They perform qualitative assessment of the usefulness or appropriateness of the risk predictions, using three datasets namely 1) a manually create a new benchmark called RiskQA, 2) emergency provider notes from Yale Medicine, and 3) sample admission notes from the MIMIC-III database. Overall the proposed approach is innovative and AgentMD introduces a new paradigm to search biomedical literature/clinical text for useful clinical risk calculator/models.

Response

We are grateful for your positive remarks on our manuscript. We have provided point-by-point responses to your concerns in the following sections.

Major Comment 1

While the proposed approach has merits as a data mining tool, it's **far from being a clinically useful tool and this point needs to be clearly emphasized in the manuscript.**

Response

Thank you for your critical comment. We believe this work introduces a novel and scalable framework for tool-augmented language agents in healthcare, and our results demonstrate its potential to support clinical reasoning and decision-making with appropriate safeguards.

Nonetheless, we agree that while AgentMD has been rigorously evaluated in our manuscript, it remains a research prototype and should be further validated before it can be incorporated into the clinical workflow. In the revision, we have emphasized this as a limitation of our study: “While AgentMD has been evaluated for both tool curation and tool usage capabilities with various datasets, it should be noted that further and more comprehensive evaluations are required before it can be incorporated into the clinical workflow.”

Major Comment 2

A key concern is that the authors don't make any attempt to assess AgentMD's **level of randomness** in risk calculator selection, signs/symptoms extraction, and risk calculations.

Response

Thank you for your thoughtful comment. We would like to clarify that we have explicitly set the temperature parameter as 0 when using large language models. Under such a setting, the language model will only generate the most likely next token in inference time. This ensures that the text generation process is greedy and thus deterministic without any randomness.

To improve clarity, we have moved this information from the Supplementary Materials to the Methods section of the revised manuscript: “We set the decoding temperature to be 0.0 across this study to ensure deterministic outputs.”

Major Comment 3

In the introduction they discuss “evidence-based risk assessment” but it’s **not clear what the level of evidence for each risk calculator is** (although they consider the number of citations as a proxy), and whether the proposed risk calculators were validated across different patient cohorts, prospectively tested, and have shown clinical utility.

Response

In the Introduction, we used the phrase “evidence-based risk assessment” to broadly describe clinical calculators, as they are generally derived from published clinical studies. However, we agree that the level of evidence supporting each calculator varies, and that it is important to provide transparency regarding their quality. In addition to the number of citations mentioned by the reviewer, we also provide other metadata that can serve as proxies for the evidence quality, including: (1) the original PubMed identifier, which the user could use to determine the publication type of the article (e.g., randomized controlled trials); (2) the size of the population studies; (3) the utility of the calculator, such as the positive predictive value.

In the revision, we have added the clarification into the discussion section: “Users can assess the quality of each curated tool using the associated publication, the population size, and the extracted utility metrics.” We have also removed the mention of “evidence-based” in the Abstract and Introduction to avoid any confusion.

Major Comment 4

Another key consideration is the **mapping of risk score input data to standardized codes** with appropriate units of measurement, etc. The generated python code does not seem to take **measurement units** as well as other factors such as the precise method of measurement or site of measurement into account (e.g., blood pressure was measured from from what body site, what method, in what position?).

Response

Thank you for your thoughtful comment. We agree that mapping input data to standardized codes and handling measurement context are important considerations when applying clinical calculators. However, many calculators require input parameters that cannot be feasibly mapped to existing standardized vocabularies. To investigate this, we analyzed the 16 clinical calculators used in our emergency department evaluation and found that the majority (11 out of 16) require at least one unstructured input parameter that lacks representation in standard terminologies such as Logical Observation Identifiers Names and Codes (LOINC). For example, the HEART score includes the criterion “positive family history (parent or sibling with CVD before age 65)”, which is not easily mapped to an ontology term.

Table. List of parameters across 16 clinical calculators. Parameters were determined as structured or unstructured using LOINC keyword search (<https://loinc.org/search>).

Calculator	Parameters	Number of parameters	Parameters that CANNOT be mapped to standard codes
Pulmonary Embolism Rule-out Criteria	age, pulse, oxygen saturation, unilateral leg swelling, hemoptysis, recent trauma or surgery, prior PE or DVT, hormone use	8	4
X-ray Need	Pain location, bone tenderness tibia, bone tenderness fibula, inability to bear weight	4	3
Canadian C-Spine Rule	age, GCS score, trauma mechanism, paraesthesia extremities, rear end collision, sitting position, ambulatory, delayed neck pain, no midline tenderness, neck rotation	10	7
Fracture Rule	age, tenderness at fibula head, isolated patella tenderness, inability to flex 90 degrees, inability to bear weight	5	4
DVT Wells' Score	active cancer, calf swelling, unilateral superficial veins, unilateral pitting edema, previous DVT, entire leg swelling, local tenderness, recent immobilization or surgery, alternative diagnosis	9	8

PE Diagnosis Score	DVT symptoms, alternative diagnosis likelihood, heart rate, immobilization or surgery, previous DVT or PE, hemoptysis, malignancy with treatment	7	5
Centor Score	temperature, cough, tonsillar swelling, anterior cervical adenopathy, age	5	2
Low Risk Score	age, glasgow score, normal mental status, scalp hematoma, loss of consciousness, injury mechanism, skull fracture, acting normally, vomiting, signs of skull fracture, severe headache	11	6
CURB-65 Mortality Score	confusion, urea, respiratory rate, systolic pressure, diastolic pressure, age	6	0
HEART Score	history, ECG abnormality, age, risk factors, troponin level	5	2
CHADS2 Index	had congestive heart failure, had hypertension, age, has diabetes, had stroke or TIA	5	0
GCS	eye score, verbal score, motor score	3	0
CT Head Rule	gcs score 15 in 2h fail, age, suspected open skull fracture, sign of basal skull fracture, vomiting episodes, amnesia before impact, dangerous mechanism of injury	7	4
qSOFA Score	systolic BP, respiratory rate, altered mentation	3	0
Glasgow Blatchford Bleeding Score	hemoglobin, BUN, systolic BP, sex, heart rate, melena present, recent syncope, hepatic disease history, cardiac failure	9	4
SIRS criteria	temperature, heart rate, respiratory rate, PaCO2, white blood cell count	5	0

With regard to units and measurement context: **Units of measurement** (e.g., mmHg for blood pressure) are typically preserved if specified in the abstract and are captured by AgentMD during tool curation. **Measurement context** (e.g., body position, site of measurement) is often absent from abstracts. In such cases, AgentMD avoids hallucinating

unspecified details and assumes general applicability unless otherwise noted. Since full-text articles might contain more granular measurement context, we acknowledge that restricting tool creation to abstracts is a limitation of our current pipeline. This is now explicitly noted in the manuscript.

In the revision, we have briefly mentioned this issue in the Introduction “the requirements of non-standardized input parameters and poor integration with electronic health records force clinicians to input data manually, interrupting clinical workflows.” We have also added the discussion that “As shown in Supplementary Table 13, many clinical calculators require non-standardized input parameters that can only be extracted from the patient notes, which necessitates the use of language agents for automatically using clinical calculators. ”

Major Comment 5

The RiskQA dataset seems like a good resource, however, it's not clear how the correct risk level was assigned to each scenario.

Response

Thank you for your positive feedback on our RiskQA dataset. We would like to clarify that the correct risk level for each scenario was manually curated based on the interpretation guidelines provided in the corresponding source abstract. For example, the CURB-65 scoring system explicitly links each score value to an associated mortality rate: a score of 0 corresponds to a mortality risk of 0.7%, a score of 1 to 3.2%, and so on. Once the ground-truth risk score was determined for a scenario, the associated risk level was assigned accordingly using these mappings.

Figure. An example abstract that contains the calculation and interpretation details of the CURB-65 score.

Multicenter Study > Thorax. 2003 May;58(5):377-82. doi: 10.1136/thorax.58.5.377.

Defining community acquired pneumonia severity on presentation to hospital: an international derivation and validation study

W S Lim¹, M M van der Eerden, R Laing, W G Boersma, N Karalus, G I Town, S A Lewis, J T Macfarlane

Affiliations + expand

PMID: 12728155 PMID: PMC1746657 DOI: 10.1136/thorax.58.5.377

Abstract

Background: In the assessment of severity in community acquired pneumonia (CAP), the modified British Thoracic Society (mBTS) rule identifies patients with severe pneumonia but not patients who might be suitable for home management. A multicentre study was conducted to derive and validate a practical severity assessment model for stratifying adults hospitalised with CAP into different management groups.

Methods: Data from three prospective studies of CAP conducted in the UK, New Zealand, and the Netherlands were combined. A derivation cohort comprising 80% of the data was used to develop the model. Prognostic variables were identified using multiple logistic regression with 30 day mortality as the outcome measure. The final model was tested against the validation cohort.

Results: 1068 patients were studied (mean age 64 years, 51.5% male, 30 day mortality 9%). Age ≥ 65 years (OR 3.5, 95% CI 1.6 to 8.0) and albumin < 30 g/dl (OR 4.7, 95% CI 2.5 to 8.7) were independently associated with mortality over and above the mBTS rule (OR 5.2, 95% CI 2.7 to 10). A six point score, one point for each of Confusion, Urea > 7 mmol/l, Respiratory rate ≥ 30 /min, low systolic (< 90 mm Hg) or diastolic (≤ 60 mm Hg) Blood pressure), age ≥ 65 years (CURB-65 score) based on information available at initial hospital assessment, enabled patients to be stratified according to increasing risk of mortality: score 0, 0.7%; score 1, 3.2%; score 2, 3%; score 3, 17%; score 4, 41.5% and score 5, 57%. The validation cohort confirmed a similar pattern.

Conclusions: A simple six point score based on confusion, urea, respiratory rate, blood pressure, and age can be used to stratify patients with CAP into different management groups.

Major Comment 6

The MIMIC-III evaluation could benefit from comparing the predicted risk against the ground truth for the top 5 conditions (or even a randomly selected set of easily obtainable outcomes). For instance, comparing the predicted risk of hospital mortality against the observed mortality and reporting AUC.

Response

Thank you for your insightful suggestion. We have conducted a new analysis evaluating whether AgentMD computation results can improve in-hospital mortality prediction, which is an important and widely studied outcome in healthcare. Specifically, we compare

AgentMD to the vanilla GPT-4 as recent studies such as [1] have demonstrated its capabilities in predicting clinical risks. Both AgentMD and GPT-4 perform zero-shot predictions in this setting. For the patient cohorts corresponding to each calculator, we draw the receiver operating characteristic (ROC) curves and computed the areas under ROC (AUC) for predicting in-hospital mortality. Among the 1,039 used ones, 604 calculators have sub-cohorts with at least one observed in-hospital death. Notably, our screening of these calculators discovered 114 useful clinical calculators curated and used by AgentMD that have higher AUC than the vanilla GPT-4. Figure 5f shows a sample of four such tools, which cover various scenarios such as high-risk varices and non-ST-elevation myocardial infarction (NSTEMI). Our results indicate that the in-hospital mortality prediction might improve with the AgentMD computation results for the patients who are eligible for these calculators.

[1] Chung P, Fong CT, Walters AM, Aghaeepour N, Yetisgen M, O'Reilly-Shah VN. Large Language Model Capabilities in Perioperative Risk Prediction and Prognostication. *JAMA Surg.* 2024;159(8):928–937. doi:10.1001/jamasurg.2024.1621

In the revision, we have added the additional analysis in the main results section and have updated Figure 5 to reflect the results.

Figure 5. Applying AgentMD on the MIMIC-III cohort. a, AgentMD is applied to 9,822 admission notes in MIMIC. b, AgentMD calculation results are aggregated by the risk calculators and patients are ranked within each tool. c, The distribution of the number of selected calculators for each patient. d, The distribution of the number of eligible patients for each used calculator. e, Calculation result distributions from two calculators. f, Sample

ROC curves of clinical calculators where AgentMD outperforms GPT-4 in predicting in-hospital mortality.

Remarks on code availability

I reviewed the LLM prompts in the appendix and they look reasonable.

Response

We appreciate your time and efforts in reviewing our manuscript and the appendix.

Responses to Reviewer #3

Overall Comment

The authors present AgentMD: an LLM-powered system to (1) curate clinical calculators from PubMed, (2) perform patient risk assessment through agent-tool use, and (3) provide population-level risk insights. The article is well written and the curation of clinical calculators at scale is novel and compelling. There are several issues, particularly in the evaluation of patient and population risk assessment (2 and 3 above), that should be addressed prior to acceptance into a high-impact journal such as Nature Communications:

Response

We are grateful for your positive remarks on our manuscript. We have provided point-by-point responses to your concerns in the following sections.

Major Comment 1

It is not clear to me why the generated calculators are only assessed on unstructured notes. I would imagine the immediate utility of these calculators would be their application to structured EHR data. For example, leveraging all structured data within the first 3 hours of presentation to the ED of Yale Medicine as opposed to only the ED provider notes.

Response

Thank you for your thoughtful comment. We agree that mapping input data to standardized codes and handling measurement context are important considerations when applying clinical calculators. However, many calculators require input parameters that cannot be feasibly mapped to existing standardized vocabularies. To investigate this, we analyzed the 16 clinical calculators used in our emergency department evaluation and found that the majority (11 out of 16) require at least one unstructured input parameter that lacks representation in standard terminologies such as Logical Observation Identifiers Names and

Codes (LOINC). For example, the HEART score includes the criterion “positive family history (parent or sibling with CVD before age 65)”, which is not easily mapped to an ontology term.

Table. List of parameters across 16 clinical calculators. Parameters were determined as structured or unstructured using LOINC keyword search (<https://loinc.org/search>).

Calculator	Parameters	Number of parameters	Parameters that CANNOT be mapped to standard codes
Pulmonary Embolism Rule-out Criteria	age, pulse, oxygen saturation, unilateral leg swelling, hemoptysis, recent trauma or surgery, prior PE or DVT, hormone use	8	4
X-ray Need	Pain location, bone tenderness tibia, bone tenderness fibula, inability to bear weight	4	3
Canadian C-Spine Rule	age, GCS score, trauma mechanism, paraesthesia extremities, rear end collision, sitting position, ambulatory, delayed neck pain, no midline tenderness, neck rotation	10	7
Fracture Rule	age, tenderness at fibula head, isolated patella tenderness, inability to flex 90 degrees, inability to bear weight	5	4
DVT Wells' Score	active cancer, calf swelling, unilateral superficial veins, unilateral pitting edema, previous DVT, entire leg swelling, local tenderness, recent immobilization or surgery, alternative diagnosis	9	8
PE Diagnosis Score	DVT symptoms, alternative diagnosis likelihood, heart rate, immobilization or surgery, previous DVT or PE, hemoptysis, malignancy with treatment	7	5
Centor Score	temperature, cough, tonsillar swelling, anterior cervical adenopathy, age	5	2
Low Risk Score	age, glasgow score, normal mental status, scalp hematoma, loss of consciousness, injury mechanism, skull fracture, acting normally, vomiting, signs of skull fracture, severe headache	11	6

CURB-65 Mortality Score	confusion, urea, respiratory rate, systolic pressure, diastolic pressure, age	6	0
HEART Score	history, ECG abnormality, age, risk factors, troponin level	5	2
CHADS2 Index	had congestive heart failure, had hypertension, age, has diabetes, had stroke or TIA	5	0
GCS	eye score, verbal score, motor score	3	0
CT Head Rule	gcs score 15 in 2h fail, age, suspected open skull fracture, sign of basal skull fracture, vomiting episodes, amnesia before impact, dangerous mechanism of injury	7	4
qSOFA Score	systolic BP, respiratory rate, altered mentation	3	0
Glasgow Blatchford Bleeding Score	hemoglobin, BUN, systolic BP, sex, heart rate, melena present, recent syncope, hepatic disease history, cardiac failure	9	4
SIRS criteria	temperature, heart rate, respiratory rate, PaCO2, white blood cell count	5	0

In the revision, we have briefly mentioned this issue in the Introduction “the requirements of non-standardized input parameters and poor integration with electronic health records force clinicians to input data manually, interrupting clinical workflows.” We have also added the discussion that “As shown in Supplementary Table 13, many clinical calculators require non-standardized input parameters that can only be extracted from the patient notes, which necessitates the use of language agents for automatically using clinical calculators. ”

Major Comment 2

I don't believe a reasonable baseline is provided to assess the benefit of tool use by the LLM. Would running all calculators (computationally cheap) and then selecting the top-5 patients by overall risk result in better or worse performance than what is presented?

Response

Thank you for your insightful comment. We would like to clarify that running *all* potential calculators on a given patient is currently impractical due to two key limitations: (1) The process of curating clinical calculators from published literature into executable code still involves significant manual effort. AgentMD, as a tool-maker, represents the first scalable framework to automate this process; (2) As mentioned in our response to your first comment, most of the clinical calculators need at least one unstructured input that can only be extracted from the clinical notes. As such, running all calculators for every patient would require extracting all possible parameters from unstructured text—an extremely challenging task.

Instead, our study compares AgentMD with a widely adopted baseline: Chain-of-Thought (CoT) prompting, which relies solely on the LLM’s internal reasoning. This allows us to isolate the benefit of tool integration. As shown in our manuscript: “AgentMD surpasses CoT by 70.1% (0.546 vs. 0.321 in accuracy, Figure 3b) and 114.4% (0.877 vs. 0.409, Figure 3c) with GPT-3.5 and GPT-4 as the base model, respectively. Surprisingly, AgentMD with GPT-3.5 even outperforms CoT with GPT-4 (0.546 vs. 0.409). These results clearly demonstrate that large language models, when provided with a well-curated toolbox of clinical calculators, can accurately select the suitable calculator and effectively perform medical calculation tasks.”

Major Comment 3

It would be beneficial if the authors could provide more objective measures of risk prediction performance.

Response

Thank you for your suggestion. We would like to clarify that most of the risk prediction evaluations are objective. Specifically, for the 500 unit tests of the curated tools (Figure 2), the AgentMD results are compared with the ground-truth results computed with online

calculator implementations (e.g., those from MDCalcs.com). In the revision, we have added 100 manually curated unit tests focused on more challenging and edge-case scenarios, as recommended by Reviewer #1. Additionally, the evaluation of AgentMD on the RiskQA dataset that contains 350 questions is also objective (Figure 3), as the questions are formulated as USMLE-style multi-choice question answering tasks. As suggested by Reviewer #2, we also screened RiskCalcs tools for their utility in predicting in-hospital mortality and identified over 100 clinical calculators curated by AgentMD that demonstrated improved predictive performance (Figure 5f).

Major Comment 4

I don't believe the Specificity, Urgency, Severity, and Absence categories generated by the LLM are properly evaluated. What is the evidence that suggests these are meaningful for population-level risk insight?

Response

Thank you for your comment. The objective of generating these scores is to quantitatively measure and rank each computed patient risk. For example, one can rank different calculator results for the same patient, as well as rank different patients for the same calculator. These scores include **Specificity** (0-100) denotes the confidence of the calculator result. Specificity is low if there are missing values and the range is wide between the risk scores of the best-case and worst-case scenarios. Specificity is high if there is no range estimation (best and worst case scenarios) and the risk calculator result contains only one specific score (not a range); **Urgency** (0-100) denotes whether the risk considered by the calculator is acute or chronic. Urgency is high if there is immediate danger worth medical attention. On the other hand, urgency is low if the risk is about 1-year or 5-year; **Severity** (0-100) denotes the extent of the calculated risk. Severity is high if the predicted risk probability is close to 100%. Severity is low if the predicted risk probability is close to 0%; **Absence** (0-100) denotes whether the calculated risk is missing in the original patient note. Absence is 100 if the calculator result (the risk) is not considered or reflected in the patient note.

Absence is 0 if the calculator result has already happened or been considered in the patient note.

To validate such scores generated by AgentMD, we have randomly sampled 30 pairs of computed patient risks. Each pair consists of the same calculator and two different patients. Two physicians were employed to annotate which patient has higher scores in each of the defined axis, with moderate to high levels of agreements: 86.7% for specificity, 61.7% for urgency, 76.7% for severity, and 66.7% for absence. On average, their annotations also align with those of AgentMD score predictions: 75.0% for specificity, 67.5% for urgency, 68.3% for severity, and 78.3% for absence. These results show that the LLM-generated scores are reasonably aligned with physician judgment and can provide a meaningful and quantitative way of assessing clinical risks.

In the revision, we have clarified the motivation for generating such scores in the main text and added the detailed definitions and additional evaluation results in the Methods section.

Remarks on code availability

The repository is well-organized and the code is well-structured and readable. I commend the authors on providing extensive source code.

Response

We appreciate your time and efforts in reviewing our manuscript and our code repository. We will maintain the codebase on a regular basis.

DEPARTMENT OF HEALTH & HUMAN SERVICES

Public Health Service

National Institutes of Health
National Library of Medicine
Bethesda, Maryland 20894

June 23, 2025

Dear Reviewers,

We sincerely thank you for the thorough evaluation and insightful comments on our manuscript titled “*AgentMD: Empowering Language Agents for Risk Prediction with Large-Scale Clinical Tool Learning*” submitted to *Nature Communications*. We deeply appreciate the time and effort you invested in reviewing our work, as well as your positive remarks and constructive feedback. Your comments have been instrumental in improving the clarity, rigor, and overall quality of the manuscript. We have carefully and systematically addressed each comment and suggestion, and we believe the revisions have significantly strengthened our work. In the following sections, we provide a point-by-point response to each of the comments raised by the reviewers, along with the corresponding changes made to the manuscript.

Zhiyong Lu, Ph.D., FACMI, FIAHSI

Senior Investigator

National Library of Medicine (NLM)

National Institutes of Health (NIH)

8600 Rockville Pike

Bethesda, MD 20894, USA

Response to Reviewer #1

Overall Comment

I've reviewed the authors' responses to my previous comments and appreciate their efforts in incorporating manually crafted test cases by domain experts, as well as providing more details about the toolset creation pipeline.

Overall, I maintain my previous positive assessment of this paper. I believe the proposed AI agent-based clinical tool represents a valuable contribution to the field, and the revised manuscript shows noticeable improvement. While I still think that relying solely on collected paper abstracts for function creation limits the method, I acknowledge that this is an area that can be improved in future work.

Response

We are grateful for your positive remarks on our revised manuscript. We also appreciate your acknowledgement that the limitation of using only the titles and abstracts can be improved in future work.

Comment on Code Availability

1. Both the created toolset (RiskCalcs) and the evaluation dataset (RiskQA) are provided in the codebase, with clear instructions on how to perform the agent's evaluation.

2. According to the author, additional details have been added regarding the toolset creation pipeline.

Response

Thank you for your time and effort in reviewing our codebase.

Responses to Reviewer #2

Major Comment 1

Thank you for your detailed response and additional data.

Overall this is an interesting paper. As I mentioned in my previous review, this work presents a novel application of LLMs and agents for data mining of published literature. However, it's important not to overstate the clinical utility of the proposed approach.

Response

We are grateful for your positive remarks on our manuscript. We also agree with you that it is important not to overstate the clinical utility of the proposed approach. Therefore, in our previous revision, we have emphasized this as a limitation of our study in the Discussions section: *“While AgentMD has been evaluated for both tool curation and tool usage capabilities with various datasets, it should be noted that further and more comprehensive evaluations are required before it can be incorporated into the clinical workflow.”*

In this revision, we have added summary sentences to the end of the Introduction section, where we have highlighted that *“Although these findings are promising, further and more comprehensive evaluation is required before its practical use in clinical settings.”*

Major Comment 2

The evaluation methods remain weak, and include certain amount of hand-picking. It's not clear to me if RiskQA is a valid dataset of risks. ChatGPT is not an appropriate baseline for mortality prediction. There are a number of other methods based on structured data that can present better baselines.

<https://github.com/YerevaNN/mimic3-benchmarks>.

Response

Thank you for your thoughtful comments. We believe ChatGPT is an appropriate baseline for two main reasons: (1) Several studies have evaluated the capabilities of ChatGPT for mortality prediction, and their results show that ChatGPT can perform this task reasonably well [1-4]; (2) ChatGPT serves as a controlled baseline for comparing with AgentMD, as their only difference is that the latter can use medical calculator tools. As such, the comparison between AgentMD and its backbone ChatGPT models can show the effectiveness of tool augmentation in the language agent. We have already cited Chung et al in the manuscript and have added the other three references in this revision.

In addition, our work mainly focuses on using language agents to predict clinical risks with free-text notes, which is already the first of its kind. Incorporating other modalities such as structured data and images is not the main focus of this paper and is worth exploration in future work. In this revision, we have acknowledged this as a limitation in the Discussions section: *“Additionally, our work focuses on risk predictions from the text modality, and incorporating other modalities such as structured data and images remains an important future direction for exploration.”*

That being said, to strengthen our evaluation, we have analyzed the effects of structured data using the suggested repository: <https://github.com/YerevaNN/mimic3-benchmarks>. Specifically, we appended the first appearance value of the processed structured data in the first 48 hours to the end of the admission note. Such structured data covers source from labs tests, vital signs, bedside monitors, and includes: capillary refill rate, diastolic blood pressure, fraction of the inspired oxygen, Glasgow coma scale (eye opening), Glasgow coma scale (motor response), Glasgow coma scale (verbal response), Glasgow coma scale total, glucose, heart rate, height, mean blood pressure, oxygen saturation, respiratory rate, systolic blood pressure, temperature, weight, and pH. We compare the differences of AgentMD’s predictions for the same patient-calculator pairs between using only the admission note and using the admission note with the appended structured data. Our new results show that: First, structured data significantly improves the specificity of AgentMD’s predictions (from 62.2% to 69.6%, paired t-test $p=2.2e-173$), which

shows that some missing input parameters in the admission notes can be compensated by using the structured data. Second, the majority of tools (54.6%) show better or the same AUC performance for mortality prediction after adding the structured data, which further indicates the potential utility of the structured data. In the revision, we have added this additional analysis to the Supplementary Materials.

[1] Chung, Philip, et al. "Large language model capabilities in perioperative risk prediction and prognostication." *JAMA surgery* 159.8 (2024): 928-937.

[2] Yoon, WonJin, et al. "LCD benchmark: long clinical document benchmark on mortality prediction for language models." *Journal of the American Medical Informatics Association* 32.2 (2025): 285-295.

[3] Han, Changho, et al. "Evaluation of GPT-4 for 10-year cardiovascular risk prediction: insights from the UK Biobank and KoGES data." *Iscience* 27.2 (2024).

[4] Oh, Namkee, et al. "ChatGPT Predicts In-Hospital All-Cause Mortality for Sepsis: In-Context Learning with the Korean Sepsis Alliance Database." *Healthcare Informatics Research* 30.3 (2024): 266-276.

Remarks on code availability

Codes are appropriate.

Response

We appreciate your time and effort in reviewing our updated code repository.

Responses to Reviewer #3

Overall Comment

I thank the authors for their thoughtful responses to my comments. However, I don't believe the revisions have sufficiently addressed the issues identified in the previous round of review:

Response

We appreciate your time and effort in reviewing our revised manuscript. We have provided point-by-point responses to your remaining concerns in the following sections.

Major Comment 1

The addition of Supplementary Table 13 supports the utility of unstructured input extraction but does not sufficiently support the omission of structured inputs. While the authors present that 68.75% of clinical calculators require at least one unstructured input parameter, they do not explore rates of data missingness. I would expect that a substantial percentage of non-missing inputs to the clinical calculators would be present in structured data but not the clinical notes. It would be helpful if the authors could provide this analysis to support their approach. Further, the highlighted challenges around non-standardized input parameters and poor integration with electronic health records, while important, are orthogonal to the omission of structured data. External labs, vital-signs, bedside monitors, etc. enter the EHR as structured data. These values are often imported into clinical notes via dotphrases, etc.

Response

Thank you for your thoughtful comments. While we have shown that 68.75% of clinical calculators require at least one unstructured input parameter in the previous revision, we agree with you that there can be input parameters that are only available in the

structured EHR data. However, our work mainly focuses on using language agents to predict clinical risks with free-text notes, which is already the first of its kind. Incorporating other modalities such as structured data and images is not the main focus of this paper and is worth exploration in future work. In this revision, we have acknowledged this as a limitation in the Discussions section: *“Additionally, our work focuses on risk predictions from the text modality, and incorporating other modalities such as structured data and images remains an important future direction for exploration.”*

In the revision, we have also conducted an additional analysis of the effects of structured data using the MIMIC-III resource suggested by Reviewer #2. Specifically, we appended the first appearance value of the processed structured data in the first 48 hours to the end of the admission note. Such structured data covers source from labs tests, vital signs, bedside monitors, and includes: capillary refill rate, diastolic blood pressure, fraction of the inspired oxygen, Glasgow coma scale (eye opening), Glasgow coma scale (motor response), Glasgow coma scale (verbal response), Glasgow coma scale total, glucose, heart rate, height, mean blood pressure, oxygen saturation, respiratory rate, systolic blood pressure, temperature, weight, and pH. We compare the differences of AgentMD’s predictions for the same patient-calculator pairs between using only the admission note and using the admission note with the appended structured data. Our results show that: First, structured data significantly improves the specificity of AgentMD’s predictions (from 62.2% to 69.6%, paired t-test $p=2.2e-173$), which shows that some missing input parameters in the admission notes can be retrieved from the structured data. Second, the majority of tools (54.6%) show better or the same AUC performance for mortality prediction after adding the structured data, which further indicates the potential utility of the structured data. In the revision, we have added this additional analysis to the Supplementary Materials.

Major Comment 2

I thank the authors for correcting my assumption that running all calculators would be computationally trivial. I would also like to clarify my previous comment regarding a

reasonable baseline to assess the benefit of tool-use by the LLM. The RiskQA experiment is cogent and CoT as a baseline supports the claim that AgentMD’s tool-use improves performance on USMLE-type questions. My concern centers around the claim that “AgentMD accurately computes medical risks with real-world patient data at an individual level.” Here, no baseline is provided to assess whether “AgentMD has the potential to assist the risk assessment process by automatically selecting and applying medical calculators to inform the risks.” To validate that the process of tool selection by AgentMD improves risk assessment, a reasonable baseline might include running all 16 calculators and presenting the top 5 patients by overall risk. I would expect this to be computationally feasible given the larger experiment on population-level risk insights.

Response

Thank you for your insightful comment. We would like to clarify that the tool selection step is necessary because medical calculators can only be applied to patients who are eligible. For example, the CURB-65 risk calculator should not be applied to patients without pneumonia, because the mortality interpretations of the score are based on pneumonia patients. Therefore, one should not run all 16 calculators on every patient. In the revision, we have added the clarification for this: *“The tool selection process is essential, as the interpretations of clinical calculators are based on the populations that participated in their original clinical studies. As such, calculators should only be applied to patients who meet the eligibility criteria.”*

That being said, we agree with the reviewer that a baseline is needed. In this revision, we have compared the tool selection performance (precision, recall, and their F1 value) of AgentMD to the baseline that uses all 16 tools as mentioned by the reviewer. Specifically, we randomly select 20 patients and annotate the calculators that they are eligible for, which is used as the ground-truth. The tool selection performance of AgentMD versus the baseline is listed below:

Method	Precision	Recall	Micro-F1
AgentMD	83.3%	69.0%	75.5%
Baseline	9.1%	100.0%	16.7%

As shown in the table, while selecting all 16 tools achieves a perfect recall, the method's precision is very low because there are only about 1.5 eligible calculators per patient. On the other hand, AgentMD's tool selection process has both high precision and recall, resulting in a much higher F1 score of 75.5% (compared to only 16.7% by the suggested baseline). Therefore, the tool selection process is critical and can be accurately performed by AgentMD.

Major Comment 3

I commend the authors on their inclusion of the in-hospital mortality prediction task which greatly strengthens the manuscript. The experiment could benefit from additional clarity in the Methods section. My understanding of the experiment, as written, is that 604 overlapping cohorts are evaluated corresponding to patients that were assigned one of 1039 calculators by AgentMD and had at least one in-hospital mortality in the cohort. Of these, AgentMD outperformed GPT-4 in 114 cohorts. It would be helpful to evaluate performance across the entire population (with the AgentMD risk score being potentially 0 for some patients).

Response

Thank you for your positive remarks on our newly added experiments. We have added more details on this screening experiment to find useful tools for in-hospital mortality prediction: *"Then, we conduct a screening study to check if any calculators can be potentially useful for in-hospital mortality prediction given the patient admission information. Specifically, we use the sum of the specificity, urgency, and severity scores to*

represent the overall risk level for each patient-calculator pair. Overall, 1,039 clinical calculators have been applied to at least one patient in the MIMIC dataset. Among these calculator cohorts, 604 have both positive and negative labels for in-hospital mortality, and we screened their corresponding tools for the in-hospital mortality prediction capability. Within the cohort eligible for each calculator, patients are ranked by their overall risk, and AUROC is used to evaluate the performance. Each tool is compared to the GPT-4 baseline which directly predicts patient mortality with chain-of-thought prompting, as prior studies have shown its effectiveness for this task.”

As stated in our response to your previous comment, we would like to clarify that the tool selection step is necessary because medical calculators can only be applied to patients who are eligible. For example, the CURB-65 risk calculator should not be applied to patients without pneumonia, because the mortality interpretations of the score are based on pneumonia patients. Therefore, one should not run all 16 calculators on every patient. In the revision, we have added the clarification for this: *“The tool selection process is essential, as the interpretations of clinical calculators are based on the populations that participated in their original clinical studies. As such, calculators should only be applied to patients who meet the eligibility criteria.”*

Major Comment 4

I again commend the authors on their validation of the Specificity, Urgency, Severity, and Absence categories generated by the LLM against physician judgement. To support the claim that “these results show that the LLM-generated scores are reasonably aligned with physician judgment and can provide a meaningful and quantitative way of assessing clinical risks,” confidence intervals should be provided so the reader can assess the significance relative to random chance.

Response

Thank you for your constructive suggestion. In the revision, we have added the confidence intervals: “*Two physicians were employed to annotate which patient has higher scores in each of the defined axis, with moderate to high levels of agreements: 86.7% (95% confidence interval (CI): 78.3–93.3%) for specificity, 61.7% (95% CI: 50.0–73.3%) for urgency, 76.7% (95% CI: 65.0–86.7%) for severity, and 66.7% (95% CI: 58.3–75.0%) for absence. On average, their annotations also align with those of AgentMD score predictions: 75.0% (95% CI: 66.7–82.5%) for specificity, 67.5% (95% CI: 59.2–75.8%) for urgency, 68.3% (95% CI: 60.0–76.7%) for severity, and 78.3% (95% CI: 70.8–85.0%) for absence. Confidence intervals are computed via bootstrapping with 10,000 times of resampling. These results show that the LLM-generated scores are reasonably aligned with physician judgment and can provide a meaningful and quantitative way of assessing clinical risks.*”

Minor Comment

The addition of the 100 manually curated unit tests strengthens the manuscript. Some of the details listed in the Results section should be in the Methods instead.

Response

We appreciate your positive remarks on our previous revision. Per your suggestion, we have moved technical details in several Results sections to the corresponding Methods sections. The changes have been tracked and highlighted in this revision.

Remarks on code availability

The repository is well-organized and the code is well-structured and readable. I commend the authors on providing extensive source code.

Response

Thank you for your positive remarks on our code repository. We will maintain the codebase on a regular basis.

Response to Reviewer #2

Overall Comment

Thank you for additional additional results with structured data. I have no further questions or comments.

Response

We are grateful for your positive remarks on our revised manuscript.

Responses to Reviewer #3

Minor Comment

I thank the authors for their thoughtful response to my comments. This will be a meaningful contribution to the field. **My only remaining minor comment** is that the physician agreement of urgency of MIMIC-III patients appears to not be significantly different than random chance (95% CI: 50.0–73.3%). I would not consider this “moderate to high levels of agreement.”.

Response

Thank you for your positive remarks and comment. In the revision, we have adjusted “moderate to high levels of agreement” to “moderate levels of agreement” to address your remaining concern.

Remarks on code availability

The repository is well-organized and the code is well-structured and readable. I commend the authors on providing extensive source code.

Response

We appreciate your time and effort in reviewing our manuscript.